# Potent BRD4 inhibitor suppresses cancer cell-macrophage interaction

Mingzhu Yin[1,2]✉, Ying Guo[1], Rui Hu[1], Wesley L. Cai[2], Yao Li[1,2], Shiyao Pei[1], Hongyin Sun[1], Cong Peng[1], Jiali Li[2], Rui Ye [2], Qiaohong Yang[3], Nenghui Wang[4], Yongguang Tao [5,6,7], Xiang Chen [1]✉ & Qin Yan [2,8,9]✉

Small molecule inhibitor of the bromodomain and extraterminal domain (BET) family proteins is a promising option for cancer treatment. However, current BET inhibitors are limited by their potency or oral bioavailability. Here we report the discovery and characterization of NHWD-870, a BET inhibitor that is more potent than three major clinical stage BET inhibitors BMS-986158, OTX-015, and GSK-525762. NHWD-870 causes tumor shrinkage or significantly suppresses tumor growth in nine xenograft or syngeneic models. In addition to its ability to downregulate c-MYC and directly inhibit tumor cell proliferation, NHWD-870 blocks the proliferation of tumor associated macrophages (TAMs) through multiple mechanisms, partly by reducing the expression and secretion of macrophage colony-stimulating factor CSF1 by tumor cells. NHWD-870 inhibits CSF1 expression through suppressing BRD4 and its target HIF1α. Taken together, these results reveal a mechanism by which BRD4 inhibition suppresses tumor growth, and support further development of NHWD-870 to treat solid tumors.

[1] Department of Dermatology, Hunan Engineering Research Center of Skin Health and Disease, Hunan Key Laboratory of Skin Cancer and Psoriasis, Xiangya Hospital, Central South University, Changsha, Hunan 410008, China. [2] Department of Pathology, Yale School of Medicine, New Haven, CT 06520, USA. [3] School of Basic Medical Sciences, Guangzhou University of Chinese Medicine, Guangzhou, Guangdong 510006, China. [4] Ningbo Wenda Pharma, Ninghai, Zhejiang 315622, China. [5] Key Laboratory of Carcinogenesis and Cancer Invasion, Ministry of Education, Xiangya Hospital, Central South University, Hunan 410078, China. [6] Key Laboratory of Carcinogenesis of Ministry of Health, Cancer Research Institute, Central South University, Changsha, Hunan 410078, China. [7] Department of Thoracic Surgery, Second Xiangya Hospital, Central South University, Changsha 410011, China. [8] Yale Cancer Center, Yale School of Medicine, New Haven, CT 06520, USA. [9] Yale Stem Cell Center, Yale School of Medicine, New Haven, CT 06520, USA. ✉email: yinmingzhu2008@126.com; chenxiangck@126.com; qin.yan@yale.edu

Epigenetic mechanisms, including post-translational modifications of histones, orchestrate chromatin organization and gene expression. As epigenetic aberrations occur frequently in cancers, targeting dysregulated epigenome has become an attractive avenue for cancer treatment[1–3]. Among the histone modifications, lysine acetylation of histone tails represents a major epigenetic mark for open chromatin and actively transcribed genes. The bromodomain and extra-terminal (BET) proteins (BRD2, BRD3, BRD4, and BRDT) are epigenetic readers that utilize tandem bromodomains (BRD) modules to recognize and dock themselves on the acetylated lysines[4–6]. These proteins play important roles in transcription activation and elongation through interaction with complexes involved in transcription including P-TEFb and mediator[7,8]. It was shown that BRD4 is highly enriched in super-enhancers that drive the expression of factors that are critical for the pathogenesis of cancer such as c-MYC, suggesting that targeting BET family proteins could be a promising approach for cancer treatment[9–12].

Dysfunction of BET proteins has been linked to the development of cancer[13,14], and compounds that selectively target BET proteins have been developed for treatment of hematological malignancies and sold tumors[5,15–17]. BET inhibitors have demonstrated remarkable efficacy in preclinical models and promising results in clinical trials of hematologic malignancies including acute myeloid leukemia (AML), acute lymphoblastic leukemia, multiple myeloma, and non-Hodgkin's lymphoma[15,18,19]. The first-generation BET bromodomain inhibitors tested in the clinic, including GSK525762 (I-BET762), OTX-015, and CPI-0610, are thienotriazolodiazepine or benzodiazepine compounds and have structures related to JQ1 (Supplementary Fig. 1). However, their effects against solid tumors in the clinic are limited, likely due to relatively low potency or various toxicities[20]. Several approaches have been utilized to improve their potency, including the development of bivalent BET inhibitors MT1[21] and BET inhibitors based on Hetero-bifunctional Proteolysis Targeting Chimera (PROTAC) technology[22–24]. Although the PROTAC compounds have shown high potency, these compounds have limited oral bioavailability. Therefore, it is essential to develop highly efficacious and orally bioavailable BET inhibitors for treatment of patients with solid tumors. Tumor-associated macrophages (TAMs) are a major type of infiltrating immune cells in various tumors[25]. Numerous studies suggested that TAMs act as a source of local and systemic cues to support the proliferation, survival, and migration of tumor cells[26,27]. It was proposed that TAMs are polarized in the tumor microenvironment toward an M2-like subtype during tumor progression, and this change underlies their ability to promote tumor growth and angiogenesis[27]. We recently reported that TAMs in peritoneal cavity promote ovarian cancer (OC) cell survival, proliferation, and early transcoelomic metastasis by facilitating TAM-OC cell spheroid formation[28,29]. In addition to their direct effects on tumor cells, TAMs suppress anti-tumor immunity by preventing activation of dendritic cells, cytotoxic T lymphocytes, and natural killer cells[25]. Thus, TAMs facilitate tumor regrowth, revascularization, spread after the treatment[30,31]. Their tumor-promoting functions are supported by clinical association of TAM infiltration with poor prognosis in most human cancer types, including ovarian cancer[32–34]. Therefore, various strategies have been developed to exploit TAMs for anti-cancer therapies[25,35]. One strategy examined in the clinic is to target colony-stimulating factor-1 receptor (CSF1R; also known as macrophage colony-stimulating factor receptor M-CSFR), which is critical for differentiation and survival of M2 TAMs.

Here we report the development and characterization of a potent BRD4 inhibitor NHWD-870. We show that NHWD-870 depletes phosphorylated BRD4 and c-MYC, and strongly suppresses the growth of multiple cancer cells derived from solid tumors. In nine mouse models of solid tumors and hematological malignancies including a patient-derived xenograft model, NHWD-870 strongly suppresses tumor growth or even induces tumor regression. NHWD-870 acts not only by directly suppressing the growth of tumor cells, but also by suppressing the proliferation of TAMs via downregulation of CSF1 in tumor cells. Mechanistically, NHWD-870 or BRD4 deletion downregulates HIF1α expression to repress CSF1 expression. These results show that BRD4 inhibition affects both cancer cells and their microenvironment and support further development of BRD4 inhibitors for treatment of solid tumors in the clinic.

## Results

**Development of a potent BRD4 bromodomain inhibitor NHWD-870.** BMS-986158 has a distinct core structure from the first-generation BET bromodomain inhibitors (Supplementary Fig. 1), and binds well with the BRD4 bromodomains. This highly selective compound exhibited strong anti-tumor affects and was tested in phase I/II clinical trial for a number of cancer indications. To further improve its efficacy and pharmacokinetic property, we first expanded the structure linked to R1 and R2 sites into oxindole and indole series (Fig. 1a). Among these compounds, the oxindole series showed similar activities against BRD4, while the methylindole compound NHWD-830 has improved potency (Fig. 1b). Further modification of these structures led to the development of the related methylindazole series (Fig. 1c). Among this series, NHWD-870 was the most potent one with a biochemical $IC_{50}$ of 2.7 nM and was chosen for further characterization (Fig. 1d and Supplementary Fig. 2). The potency of NHWD methylindazole series was also reflected in cellular assays using NCI-H211 small cell lung cancer (SCLC) cells (Fig. 1e) and MDA-MB231 triple negative breast cancer (TNBC) cells (Fig. 1f). NHWD-870 ($IC_{50} = 2$ or 1.6 nM) showed about three-fold increased potency from BMS-986158 ($IC_{50} = 6.6$ or 5 nM) and is about 50-fold more potent than JQ1 ($IC_{50} = 102$ or 65 nM) (Fig. 1e, f). We also compared NHWD-870 with three other potent BET inhibitors in cellular assays using A375 melanoma cells (Fig. 1g). NHWD-870 ($IC_{50} = 2.46$ nM) is about 23-fold more potent than I-BET151 ($IC_{50} = 55.5$ nM), and about 14-fold more potent than two clinical stage inhibitors GSK-525762 ($IC_{50} = 35.6$ nM) or OTX015 ($IC_{50} = 34.8$ nM) (Fig. 1g). These results indicated that NHWD-870 is a very potent BRD4 inhibitor.

**NHWD-870 suppresses the growth of multiple solid tumor cell lines.** We then compared NHWD-870 with JQ1 for their cellular effects in a series of cancer cell assays. JQ1 and other BET inhibitors was shown to suppress cell growth partly through down-regulation of c-MYC transcription and it was shown recently that phosphorylation of BRD4 by CK2, which is required for its activation and contribute to resistance to JQ1[36,37]. RT-qPCR and western blot analysis of a panel of solid tumor cell lines showed that BRD4 is highly expressed in H526 SCLC, A2780 and ES-2 ovarian cancer, MDA-MB231 TNBC cells, and A375 melanoma cells (Supplementary Fig. 3). Treatment of these solid tumor cell lines with 10 nM NHWD-870 led to the depletion of phosphorylated BRD4 and c-MYC, while even 50 nM JQ1 had no effect (Fig. 2a). Unlike the PROTAC compounds, NHWD-870 had minimal effects on the stability of BRD4 (Fig. 2a).

In cell growth assays, compared with the modest effects of JQ1 on H526, A2780, ES-2, and MDA-MB231 cells, NHWD-870 showed strong inhibitory activities against these cells in 5-day assays (Fig. 2b), but had minimal effects on immortalized human keratinocytes HaCAT cells (Supplementary Fig. 4).

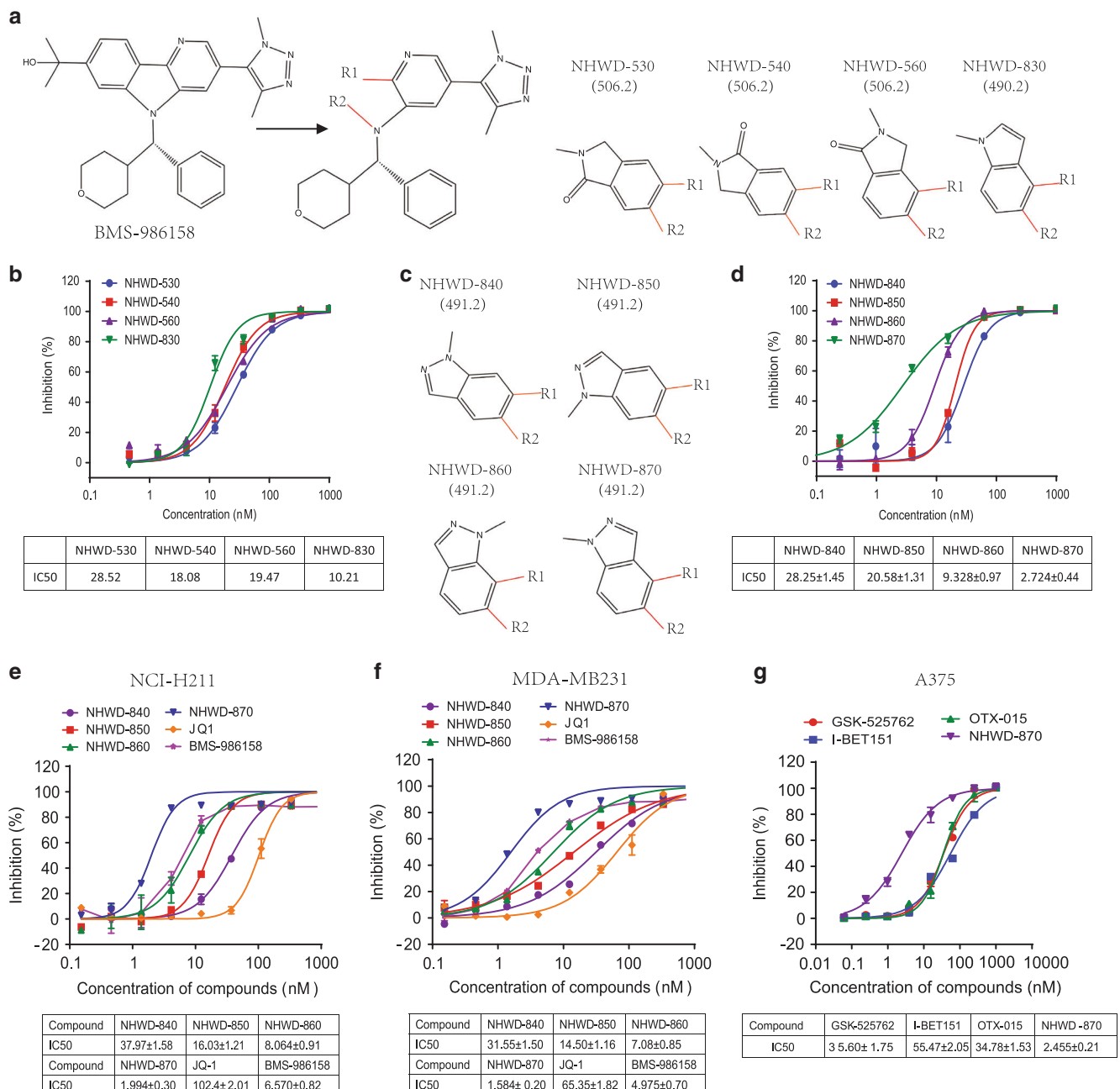

**Fig. 1 Rational design of a potent BET inhibitor NHWD-870. a** Chemical structure of BET inhibitor BMS-986158 and its analogs NHWD-530, NHWD-540, NHWD-560, and NHWD-830. **b** Biochemical activity of NHWD-530, NHWD-540, NHWD-560, and NHWD-830 against BRD4 (BD1 + BD2). The bottom panel shows the average IC$_{50}$ values for these compounds. **c** Chemical structures of NHWD-830 analogs NHWD-840, NHWD-850, NHWD-860, and NHWD-870. **d** Biochemical activity of NHWD-840, NHWD-850, NHWD-860, and NHWD-870 against BRD4 (BD1 + BD2). Bottom panel shows the average IC$_{50}$ values for these compounds. **e**, **f** Cellular activities of NHWD-840, NHWD-850, NHWD-860, NHWD-870, JQ-1, and BMS-986158 in SCLCs (H211) (**e**) and TNBCs (MDA-MB231) (**f**), as measured by alamarBlue assays. Bottom panels show the average IC50 values for these compounds. **g** Cellular activities of GSK-525762, I-BET151, OTX-015, and NHWD-870 in melanoma cells (A375), as measured by alamarBlue assays. The bottom panels show the average IC50 values for these compounds. Three independent experiments were performed for all experiments. Data are plotted as mean ± SEM from two independent experiments. Source data are provided as a Source Data file.

NHWD-870 strongly suppressed the ability of these cells to form colonies, while JQ1 only exhibited modest effects at the same concentration (Fig. 2c, d). Furthermore, in anchorage-independent 3D matrigel assays, NHWD-870 strongly reduced the colony size while JQ1 had minimal effects (Fig. 2e, f), further supporting the higher potency of NHWD-870 than JQ1. These results are consistent with the comparison of NHWD-870 with JQ1 and four other potent BET inhibitors in other cellular assays

(Fig. 1e-g). We then asked whether BRD4 depletion has similar effects as NHWD-870 in these assays. BRD4 depletion in A375 melanoma cells using the CRISPR/Cas9 system significantly decreased the expression of c-MYC and suppressed the ability of these cells to form colonies (Supplementary Fig. 5). These results are consistent with the notion that BRD4 inhibition by NHWD-870 mediates the decrease of c-MYC expression and cell growth in vitro.

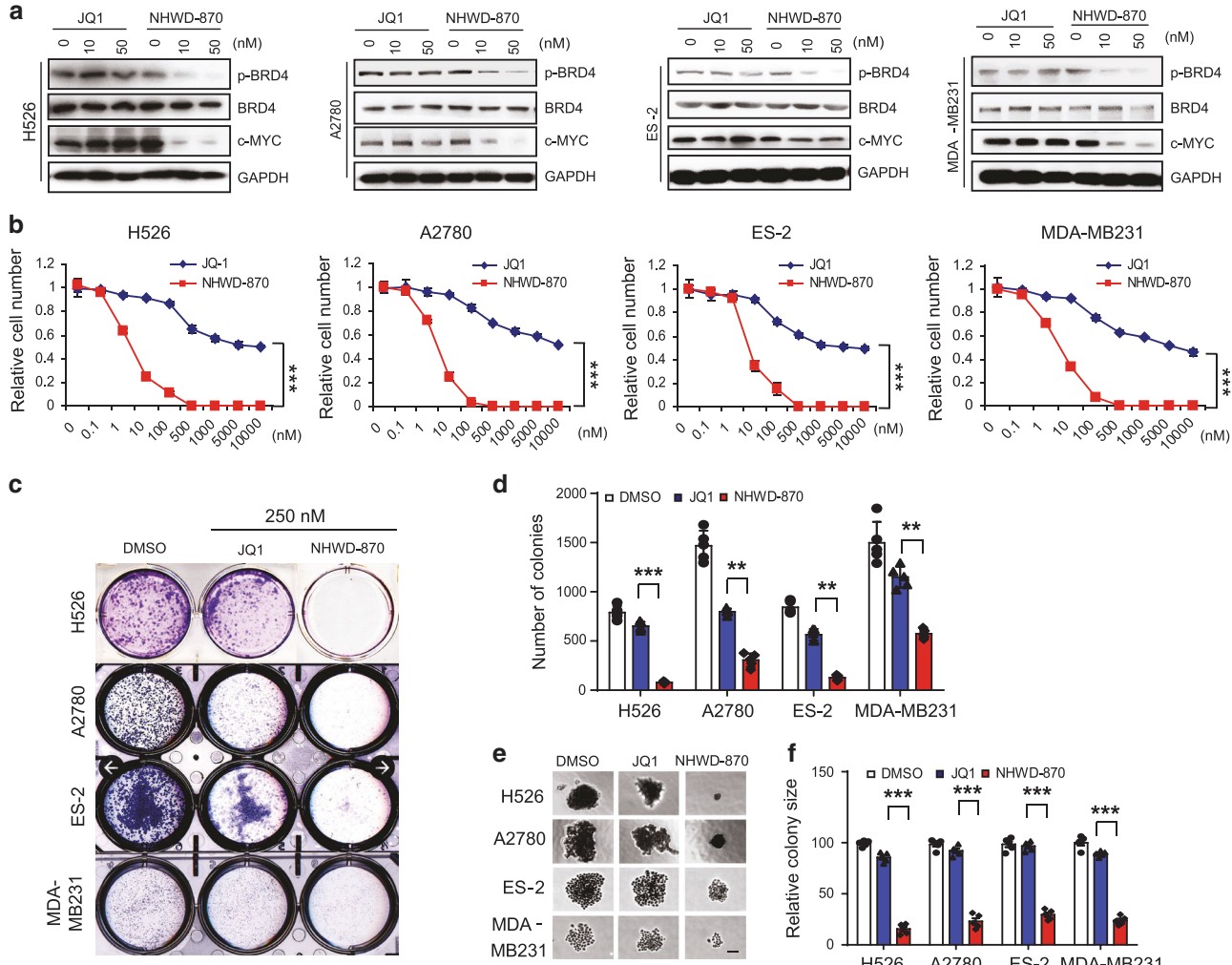

**Fig. 2 NHWD-870 inhibited BRD4 phosphorylation and c-MYC expression, and suppressed cell growth. a** Representative western blot analysis of p-BRD4 (s484/488), BRD4, c-MYC, and GAPDH after treatment of MDA-MB231, SUM159, A2780 ES-2, and H526 cells with JQ-1 or NHWD-870 at the indicated concentrations for 24 h. GAPDH served as the loading control. Selected from three independent experiments. **b** MTT assays of H526, A2780, ES-2, and MDA-MB231 cells treated with JQ-1 or NHWD-870 at the indicated concentrations for 5 days. Data are presented as mean ± SEM from three independent experiments. **c, d** Clonogenic assays of H526, A2780, ES-2 and MDA-MB231 cells treated with 250 nM JQ-1 or NHWD-870 for 5 days. Shown are representative images (**c**) and quantification of colonies (**d**). Data are presented as mean ± SEM from five independent experiments. **e, f** 3D matrigel assays of H526, A2780, ES-2, and MDA-MB231 cells treated with 100 nM JQ-1 or NHWD-870. Shown are representative images (**e**) and quantification of relative colony sizes (**f**). Scale bar, 50 μm. Data are presented as mean ± SEM from five independent experiments. *p* values were calculated using two-tailed, unpaired *t* tests in this figure. **$p < 0.01$; ***$p < 0.001$. Source data are provided as a Source Data file.

**NHWD-870 exhibits drug-like properties for clinical development**. It was reported that JQ1 had a short half-life of 7, 13, and 13 min when incubated with mouse, rat, and human liver microsomes, respectively[38]. NHWD-870 showed more stable metabolic profile when incubated with mouse, rat, dog, monkey, and human liver microsomes, with the half-life ranging from 21 to 39 min (Supplementary Fig. 6 and Supplementary Table 1). Furthermore, equilibrium dialysis assays showed that NHWD-870 exhibited moderate binding rate with plasma protein and good recovery rate, which are comparable to Warfarin, an anticoagulant drug used in the clinic (Supplementary Table 2). Taken together, these results suggested that NHWD-870 has acceptable metabolic profile.

The strong cellular potency and acceptable in vitro metabolic profile prompted us to examine its pharmacokinetic profiles. We first tested the oral absorption, metabolism, and bioavailability of NHWD-870 in mouse and rat. Both intravenous and oral administration of NHWD-870 showed acceptable absorption

rate into the circulation (Supplementary Fig. 7a, b). We then evaluated NHWD-870 for its in vivo metabolic profile and systemic exposure in mouse and rat. A range of clearance rates and exposures were observed, with oral bioavailability generally favorable for NHWD-870 (Supplementary Table 3). Based on the above data, we then evaluated the concentration of NHWD-870 in the H526 xenograft model. Orally administrated NHWD-870 showed good penetration into the lungs and SCLC tumors, comparable to BMS-981658 at the same concentration (Supplementary Fig. 7c, d), suggesting that NHWD-870 is a potential drug candidate for the treatment of solid tumors.

One major issue with current BET inhibitors is their toxicities. It was reported that some BET inhibitors induced cardiotoxicity, we, therefore, assessed the cardiotoxicity of NHWD-870 using hERG safety assays. NHWD-870 exhibited mild inhibition of hERG channel (IC$_{50}$ = 5.4 μM) (Supplementary Fig. 8a). In comparison, the positive control Dofetilide, used to treat abnormal heartbeats, strongly inhibited hERG

channel ($IC_{50} = 0.014\ \mu M$, consistent with reported values) (Supplementary Fig. 8b). Previous animal studies of many BET inhibitors appeared to show various degree of body weight loss[39,40], commonly used as a surrogate for gastrointestinal toxicity for drug treatment in animal models. To this end, we examined the effects of NHWD-870 and BMS-986158 on body weight changes and found that NHWD-870 caused about 6% body weight loss, which is considered acceptable and much less severe than BMS-986158, which led to about 11% body weight loss (Supplementary Fig. 8c and Supplementary Table 4). Thus,

the high potency and tolerable toxicity of NHWD-870 indicated that it is a strong candidate for further clinical development.

**NHWD-870 has strong anti-tumor activities in nine mouse models.** We next examined the effects of NHWD-870 on tumor growth in nine different mouse models. Dose titration studies showed that oral administration of NHWD-870 showed dosage-dependent effects on H526 SCLC tumors (Fig. 3a). Remarkably, at the relative low dosage of 3 mg/kg per day or 1.5 mg/kg twice

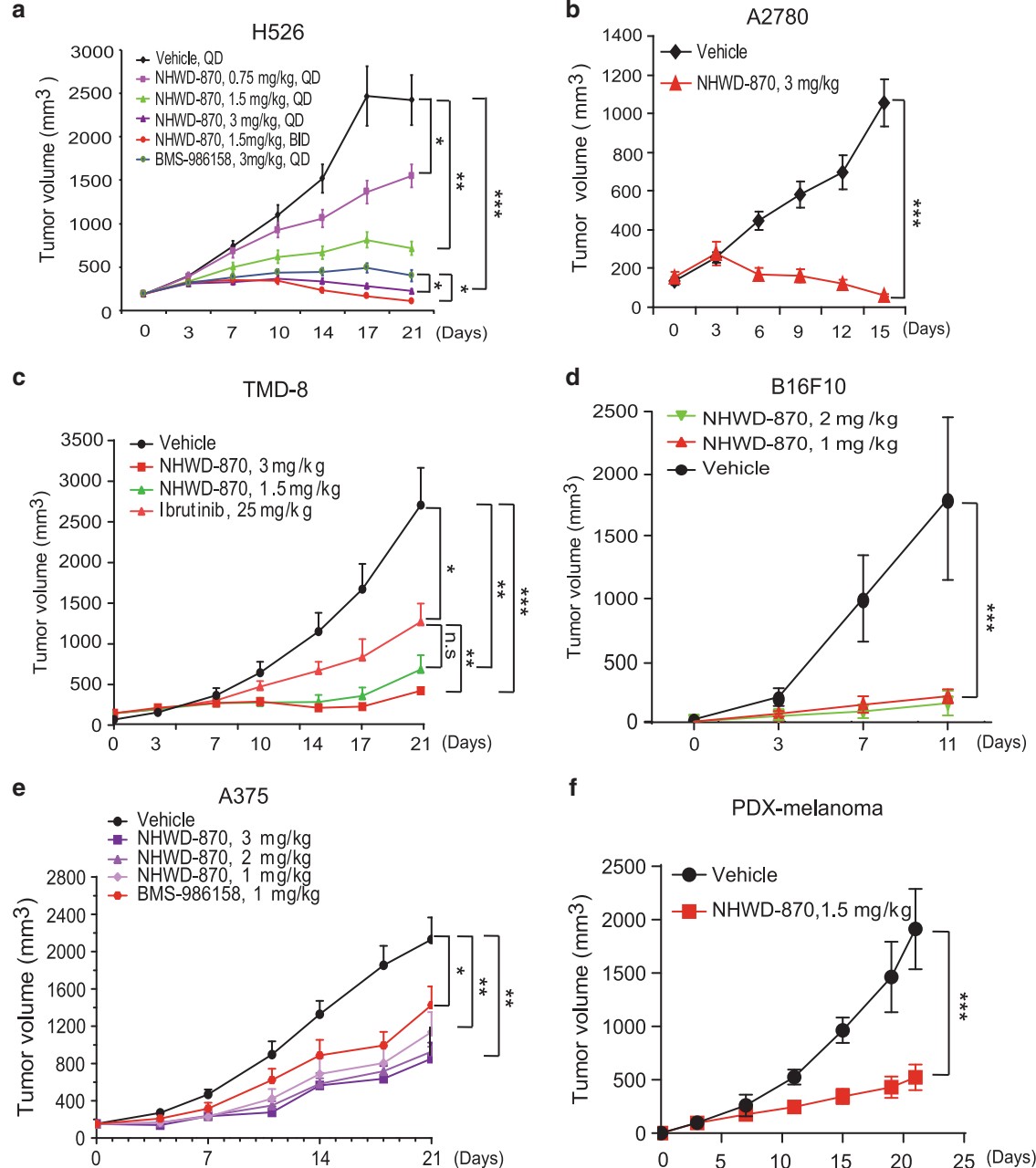

**Fig. 3 Oral administration of NHWD-870 strongly suppressed the growth of established lung tumor, ovarian tumor, lymphoma, and melanoma in vivo. a** Tumor growth curves for H526 SCLC tumor-bearing mice ($n = 6$) treated with the indicated compounds for 21 days. PO, oral administration; QD, once daily; BID, twice daily. **b–d** Tumor growth curves for A2780 ovarian tumor ($n = 5$) (**b**), TMD8 large B cell lymphoma ($n = 6$) (**c**), and B16F10 melanoma ($n = 5$) (**d**) bearing mice treated with the indicated compounds once daily for 11–21 days. **e, f** Tumor growth curves for A375 melanoma ($n = 6$) (**e**), and patient-derived xenograft (PDX) of melanoma ($n = 5$) (**f**) bearing mice treated with the indicated compounds once daily (5 days on, 2 days off) for 21 days. For **d–f**, NHWD-870 used is NHWD-870-HCl synthesized using Suzuki coupling route. Data are presented as mean ± SEM in (**a**, **b**, **d**, **f**) and mean + SEM in (**c**, **e**). $p$ values were calculated using two-tailed, unpaired $t$ tests. *$p < 0.05$; **$p < 0.01$; ***$p < 0.001$. Source data are provided as a Source Data file.

per day, NHWD-870 regressed established H526 SCLC tumors, with slightly better effects than BMS-986158 (Fig. 3a and Supplementary Fig. 9a). Its strong tumor regression activity was also observed with A2780 ovarian tumors (Fig. 3b and Supplementary Fig. 9b). NHWD-870 showed stronger tumor-suppressive effects than Bruton's tyrosine kinase (BTK) inhibitor Ibrutinib in TMD8 large B cell lymphoma model (Fig. 3c). Consistently, NHWD-870 also showed very strong tumor inhibitive effects on B16F10 mouse melanoma, A375 human melanoma and a human melanoma patient-derived xenograft (PDX) model (Fig. 3d–f). Its effects were slightly better than the effects of BMS-986158 in A375 model (Fig. 3e). It also showed strong tumor-suppressive effects on ES-2 ovarian cancer and MDA-MB231 TNBC models (Supplementary Fig. 9c, d). To begin to examine the mechanisms into its tumor-suppressive activities, we examined its effects on apoptosis and proliferation using cleaved caspase 3 and Ki67 staining. NHWD-870 showed robust activities inducing apoptosis and suppressing cell proliferation, with stronger effects than BMS-986158 (Supplementary Fig. 10a–d), suggesting that this potent and orally bioavailable BRD4 inhibitor could be used for diverse indications against solid tumors and lymphoma in the clinic.

**NHWD-870 reduces the proliferation of TAMs in mouse models**. To further understand the mechanisms by which BRD4 inhibition affected tumor growth, we examined the effects of NHWD-870 treatment on tumor microenvironment. As we recently reported that in ovarian cancer, TAMs form spheroid with ovarian cancer cells, facilitate their implantation, and promote tumor cell proliferation and transcoelomic metastasis[28,29], we focused our efforts on TAMs. We observed that the numbers of TAMs (CD68[+] cells) were significantly lower in subcutaneous tumors formed by either H526 SCLC or A2780 OC cells after NHWD-870 treatment (Fig. 4a, b). Moreover, proliferating TAMs (CD68[+]Ki67[+] cells) significantly decreased in NHWD-870 treated A2780 tumors (Fig. 4c, d, Supplementary Fig. 10e).

To extend our findings, we examined the effects of NHWD-870 in an orthotopic mouse OC model, in which we intraperitoneally injected A2780 OC cells, alone or together with F4/80[+]CD11b[+]CD206[+] TAMs isolated from the tumor tissues of OC-bearing donor mice 3 days later (Fig. 4e). As we reported previously[29], supplement of TAMs markedly increased tumor burden, as indicated by tumor weight and the amount of ascitic fluid (Fig. 4f, g, Ctrl+TAM vs Ctrl). These phenotypes were due to increased proliferation of both tumor cells and macrophages (Ki67[+]CD68[−] and Ki67[+]CD68[+] cells, respectively) (Fig. 4h–i, Ctrl+TAM vs Ctrl). NHWD-870 significantly reduced tumor weight and the volume of ascitic fluid, as well as the percentage of proliferating tumor cells and macrophages (Fig. 4f–j, NHWD-870 vs Ctrl). NHWD-870 also decreased the number of TAMs (CD11b[+]F4/80[+] cells) or proliferating TAMs (CD11b[+]F4/80[+]CD68[+]Ki67[+] cells) in A375 melanoma model (Supplementary Fig. 11a, b). Surprisingly, addition of exogenous TAMs was unable to rescue the strong effects of BRD4 inhibition on tumor cell proliferation and tumor growth (Fig. 4f–i, NHWD-870+TAM vs TAM and Ctrl). The inability to rescue was not due to the limited amount of TAMs (Fig. 4k, CD68[+] cells and Supplementary Fig.11c, d, CD11b[+]CD206[+] cells, NHWD-870+TAM vs Ctrl), but likely due to the limited number of proliferating TAMs (Fig. 4j, Ki67[+]CD68[+] cells, NHWD-870+TAM vs Ctrl). These results showed that NHWD-870 significantly inhibited the proliferation of TAMs.

**NHWD-870 inhibits TAM proliferation by inhibiting CSF1R signaling**. Direct treatment of CD11b[+]CD206[+] TAMs isolated

from A2780 ovarian cancer-bearing donor mice with 100 nM NHWD-870 led to <50% decrease of the ability of these cells to form colonies in a 3D culture system (Supplementary Fig. 11e, f). This effect is much smaller than the almost complete disappearance of proliferating TAMs after NHWD-870 treatment (94% inhibition) in mouse OC model [Fig. 4j, MHWD870 (0.16%) vs Ctrl (2.6%)]. This observation prompted us to ask whether the robust inhibitory effect of NHWD-870 on TAMs was mainly due to the inability of NHWD-870-treated tumor cells to support the proliferation of TAMs. To this end, we turned to tumor cell and TAMs co-culture systems. In the first set of experiments, A2780 cells were pretreated with 100 nM NHWD-870 or mock treated before directly co-cultured with TAMs. Pretreatment of A2780 cells with NHWD-870 led to significant decrease of the proliferation rate of TAMs (83% inhibition) (Fig. 5a, b). In the second set of experiments, we asked whether this decrease is due to secreted factors from 100 nM NHWD-870 treated tumor cells. Co-culturing of A2780 cells with TAMs in separate chambers (Fig. 5c) led to ~8 fold increase of the size of colonies formed by TAMs in a 3D co-culture system (Fig. 5d, e). Pretreatment of A2780 cells with NHWD-870 completely abolished the ability of A2780 to support the growth of TAMs (Fig. 5d, e). Consistently, pretreatment of A2780 cells with NHWD-870 potently decreased the growth of TAMs in a dose-dependent manner in a similar 2D co-culture system (Supplementary Fig. 11g–i). Taken together, these data suggested that NHWD-870 inhibited TAM proliferation mainly through affecting secreted factor(s) from tumor cells.

To define the molecular mechanisms by which NHWD-870 inhibited proliferation of TAMs, we examined the mRNA levels of various growth factors and chemokines, including EGF, FGF2, HGF, IGF-1, PDGF-A, PDGF-B, PIGF, TGF-β, TNF-α, VEGF-A, VEGF-B, VEGF-C, CCL5, CCL17, CCL18, CXCL5, CXCL8, CXCL9, CXCL13, and CSF1 in ID8 mouse ovarian cancer cells, B16 mouse melanoma cells, TAMs and monocytes by RT-qPCR. We found that *Csf1* was highly expressed in ID8 and B16 tumor cells (Supplementary Fig. 12a, b). However, *Csf1* is expressed at very low level in TAMs and monocytes (Supplementary Fig. 12a, b). In contrast, *Csf1r*, the gene encoding CSF1 receptor, is highly expressed in TAMs and monocytes, but not in tumor cells (Supplementary Fig. 12a, b). We then asked whether NHWD-870 affected *CSF1* expression in tumor cells. RT-qPCR analyses showed that NHWD-870 treatment significantly decreased *CSF1* mRNA in a panel of moue and human ovarian cancer and melanoma cell lines (Fig. 5f). Moreover, ELISA analysis indicated that NHWD-870 blocked CSF1 secretion by ID8, B16, A2780, SKOV3, and A375 cells (Fig. 5g). Consistently, immunofluorescence staining showed that NHWD-870 blocked CSF1 expression in A2780 cells in vitro (Fig. 5h, i) and A2780 tumors in vivo (Fig. 5j, k).

As CSF1/CSF1R signaling was shown to promote the proliferation of TAMs[25,35], we asked whether the CSF1/CSF1R signaling pathway mediated the inhibitory effects of NHWD-870 on proliferation of TAMs. CSF1 increased proliferation of TAMs in a dose-dependent manner (Fig. 6a). While pretreatment of A2780 cells with NHWD-870 strongly affected the ability of A2780 cells to support the proliferation of TAMs in a separate chamber in co-culture assays (Fig. 5c–e and Fig. 6b, c), 10 ng/ml CSF1 was able to partially rescue the inhibitory effects of NHWD-870 (Fig. 6b, c), suggesting other mechanisms, together with CSF1 downregulation in tumor cells, contribute to indirect effects of NHWD-870 on macrophages. Consistent with these results, treatment of TAMs with CSF1R neutralization antibody inhibited proliferation of TAMs to the similar extent as NHWD-870 pretreated of A2780 cells (Fig. 6d). Furthermore, there is no synergistic effects by combining treatment with CSF1R neutralization antibody and co-culturing with NHWD-870 pre-treated

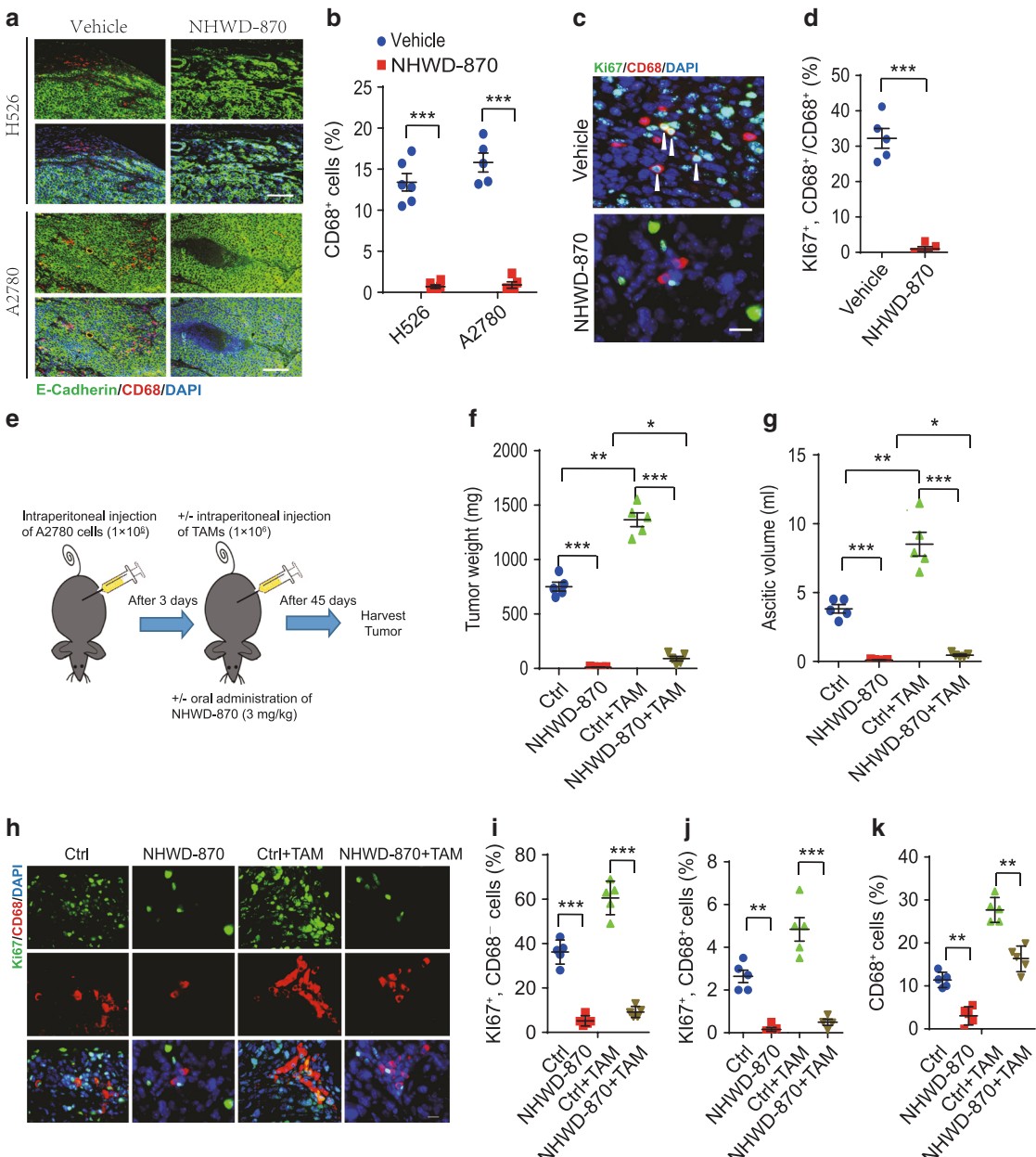

**Fig. 4 NHWD-870 reduced the proliferation of TAMs in vivo. a, b** NHWD-870 treatment reduced the number of TAMs in subcutaneously implanted H526 and A2780 tumors. Shown are immunofluorescent staining of CD68 (macrophage marker, red) and E-Cadherin (cancer cells, green) (**a**) and quantification of CD68 positive cells (**b**) in vehicle, NHWD-870 treated H526 ($n = 6$) and A2780 ($n = 5$) tumors from Fig. 3a, b, respectively. Scale bar is 100 μm. Data are presented as mean ± SEM from different tumors. **c, d** NHWD-870 treatment reduced the proliferation TAMs in subcutaneously implanted A2780 tumors. Shown are representative immunofluorescent staining of Ki67 and CD68 (**c**) and quantification of percentage of Ki67+ cells in CD68+ cells (**d**). Scale bar is 30 μm. Data are presented as mean ± SEM of five different tumors. **e** Schematics of the animal experiments using an orthotopic mouse OC model, established by intraperitoneally injecting human A2780 cells into female recipient nude mice. Mice were then either untreated (Ctrl) or treated with NHWD-870 (PO, QD, 3 mg/kg) for 45 days. Half of the recipient mice received A2780 cells plus TAMs isolated from tumors of OC-bearing donor mice. **f, g** Tumor weights (**f**) and ascitic fluid volumes (**g**) at day 45 for mice in (**e**). Data are presented as mean ± SEM from five different mice. **h–k** NHWD-870 treatment decreased proliferation of the both tumor cells and TAMs. Shown are representative immunofluorescent staining of Ki67 and CD68 (**h**) and quantification of Ki67+CD68− (**i**) or Ki67+CD68+ (**j**), or CD68+ cells (**k**). Scale bar is 50 μm. Data are presented as mean ± SEM from five different mice. $p$ values were calculated using two-tailed, unpaired $t$ tests in this figure. *$p < 0.05$; **$p < 0.01$; ***$p < 0.001$. Source data are provided as a Source Data file.

A2780 cells (Fig. 6b, d), suggesting that these treatments act through the same pathway.

Treatment of TAMs with CSF1 induced PI3K/AKT1 and ERK phosphorylation (Supplementary Fig. 13, CSF1 vs Ctrl), and these phosphorylation events were not affected by direct treatment of TAMs with NHWD-870 (Supplementary Fig. 13, CSF1 +

NHWD-870 vs CSF1). Co-culturing of TAMs with A2780 cells in separate chambers induced PI3K/AKT1 and ERK activation in TAMs (Supplementary Fig. 13, A2780 vs Ctrl), and their activation was blocked by pre-treatment of A2780 cells with NHWD-870 (Supplementary Fig. 13, A2780 + NHWD-870 vs A2780). These results suggest that NHWD-870 inhibits CSF1

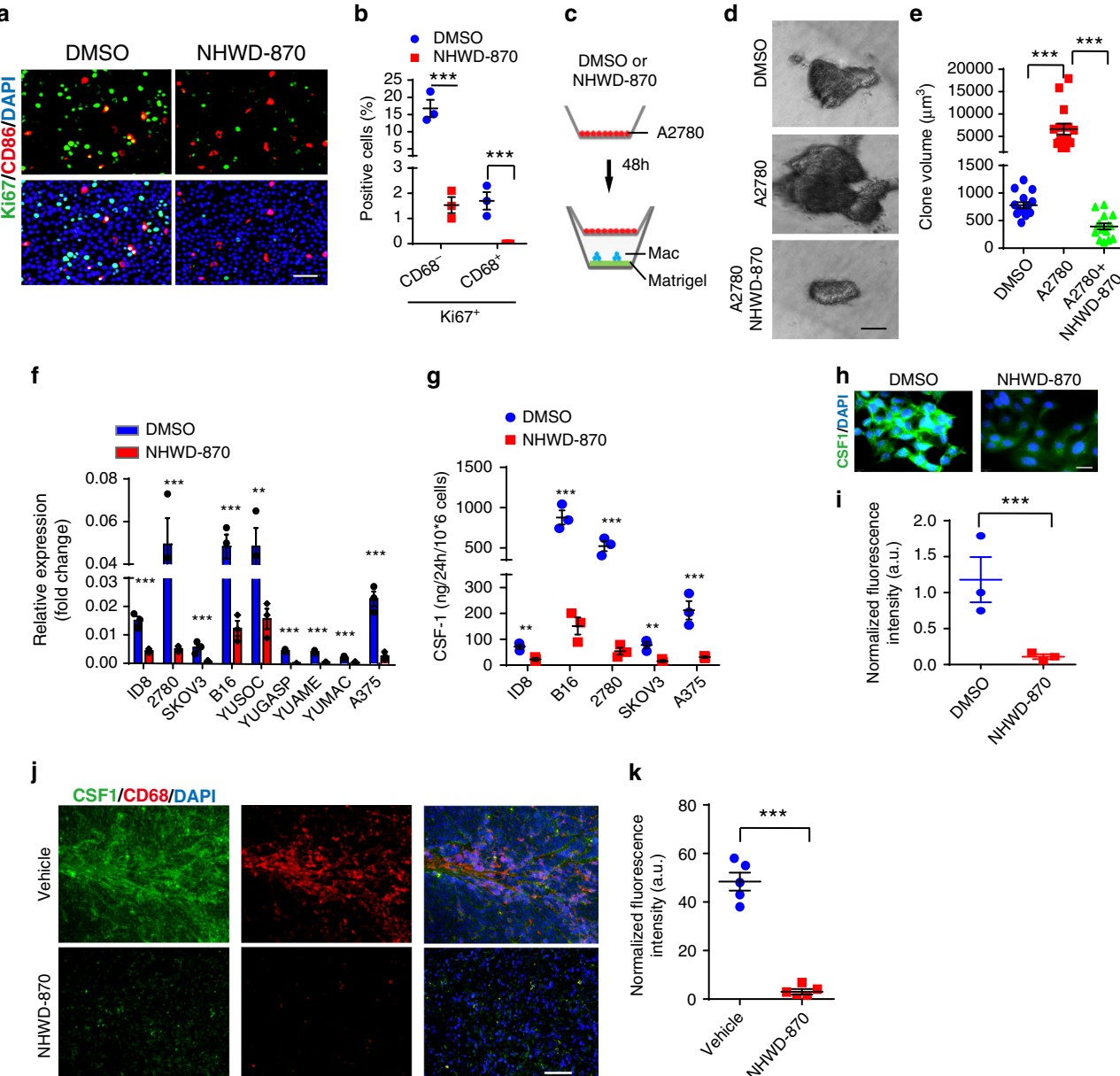

**Fig. 5 NHWD-870 downregulated CSF1 expression in tumor cells to inhibit TAM proliferation. a, b** NHWD-870 inhibited tumor cell and TAM proliferation in vitro. A2780 tumor cells were pretreated with 100 nM NHWD-870 and then directly co-cultured with TAMs (A2780:TAMs ratio as 4:1) at the density of $1 \times 10^5$ total cells in a 12-well. Shown are immunofluorescent staining of Ki67 and CD68 (**a**) and quantification of Ki67+ and CD68− or CD68+ cells (**b**). Scale bar is 50 μm. Data are presented as mean ± SEM from three independent experiments. **c–e** NHWD-870 significantly inhibited tumor cell supported TAM proliferation. A2780 cells (pre-treated with or without 100 nM NHWD-870 for 48 h) were seeded into the top chamber (transwell size: 0.4 μm) and TAMs (Mac, 40,000 cells per 24-well) in medium containing 2% matrigel were seeded into the bottom chamber pre-coated with Matrigel. No NHWD-870 was added to the co-culture. The upper chamber without A2780 cells was used as a control. Shown are schematics of transwell co-culture experiment (**c**), representative pictures of TAM spheroid (**d**: black/white), and quantification of spheroid volumes (**e**). Scale bar is 50 μm. Data are presented as mean ± SEM from 15 independent experiments. **f** RT-qPCR analyses of relative *CSF1* mRNA level in ovarian cancer cells (ID8, A2780, SKOV3, and ES-2) and melanoma cells (B16, YUSOC, YUGASP, YUAME, YUMAC, and A375) treated with 50 nM NHWD-870 for 48 h. Data are presented as mean ± SEM from three independent experiments. **g** CSF1 protein levels in supernatant of $10^6$ ID8, B16, A2780, SKOV3, and A375 cells treated with DMSO or 25 nM NHWD-870 for 24 h, as measured by ELISA. Data are presented as mean ± SEM from three independent experiments. **h** Representative immunofluorescent staining of CSF1 in DMSO or 100 nM NHWD-870 treated A2780 cells. Scale bar is 20 μm. **i** Quantification of CSF1 staining in DMSO or 100 nM NHWD-870 treated A2780 cells. Data are presented as mean ± SEM from three independent experiments (6 fields per sample). **j, k** NHWD-870 significantly reduced CSF1 expression in tumors from A2780 tumor-bearing mice treated with vehicle or NHWD-870 for 15 days. Shown are representative immunofluorescent staining of CSF1 (green) and CD68 (red) (**j**) and quantification of CSF1 staining (**k**). Data are presented as mean ± SEM from five different tumors. *p* values were calculated using two-tailed, unpaired *t* tests in this figure. **p < 0.01; ***p < 0.001. Source data are provided as a Source Data file.

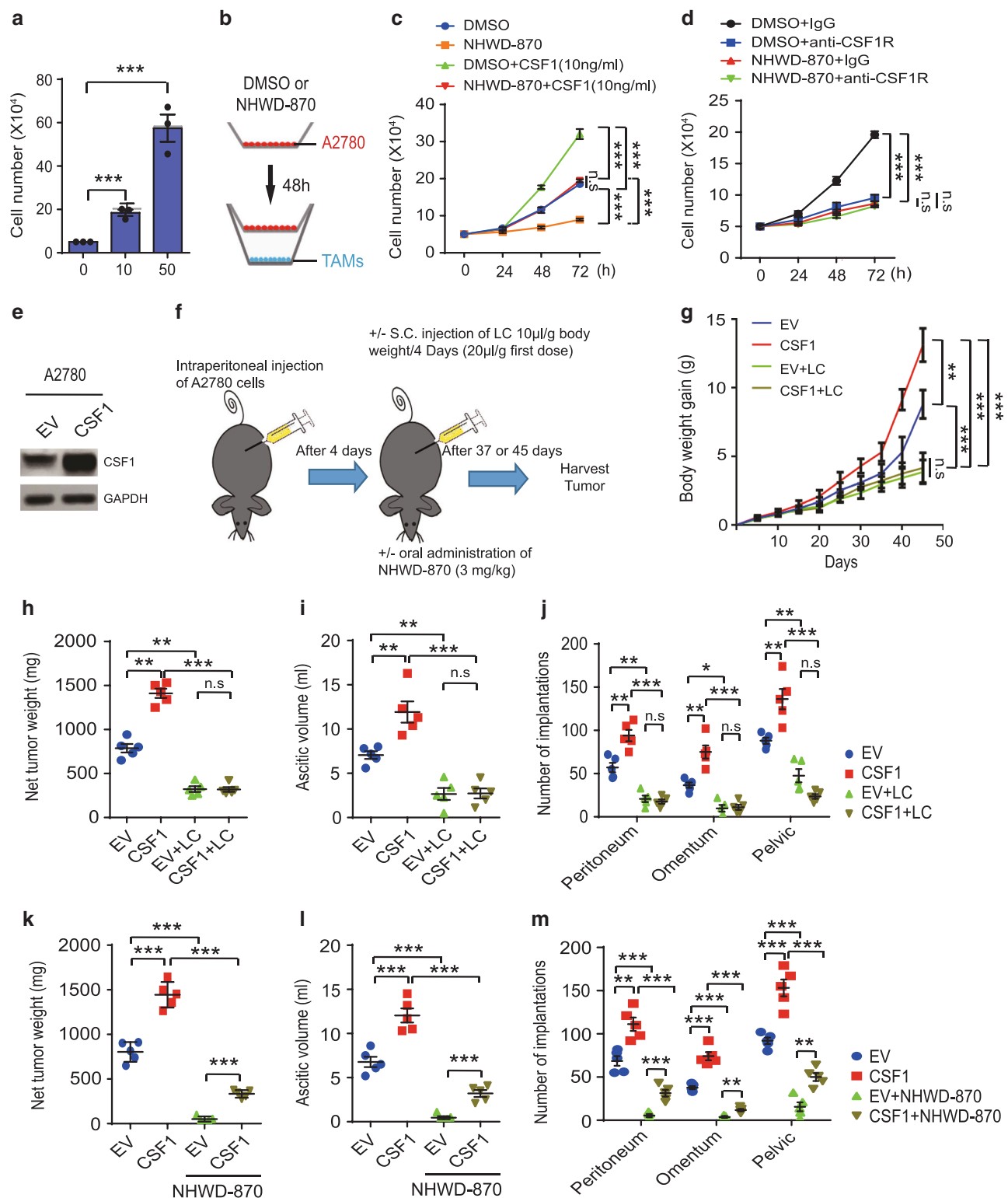

expression in tumor cells, which leads to reduced CSF1/CSF1R signaling in TAMs and decreased proliferation rate of TAMs. To determine whether BRD4 directly regulates TAM proliferation, we evaluated the effects of NHWD-870 treatment and CRISPR/Cas9-mediated BRD4 knockout (KO) on TAMs induced by CSF1. Both NHWD-870 treatment and BRD4 deletion decreased TAMs proliferation by ~50% as measured by Ki67 FACS (Supplementary Fig. 14), suggesting that NHWD-870 also has some direct anti-proliferative effects on macrophages.

We next asked whether CSF1 promotes tumor formation through TAMs in A2780 orthotopic tumor model. CSF1 was stably transfected into A2780 cells (Fig. 6e), and the control and CSF1 overexpressing cells were intraperitoneally injected into nude mice (Fig. 6f). To determine the contribution of macrophages, a group of the tumor-bearing mice were treated with liposomal clodronate (LC) to deplete macrophages as done previously[41], and another group of mice were treated with NHWD-870 (Fig. 6f). CSF1 overexpression significantly increased

**Fig. 6 CSF1/CSF1R signaling is critical for the tumor inhibitory effects of NHWD-870 and CSF1 induced ovarian cancer growth depends on macrophages. a** Number of TAMs after treatment with the indicated concentration of CSF1 for 48 h. Data are presented as mean ± SEM from three independent experiments. **b–d** Schematics of the experiments (**b**). A2780 cells (pre-treated with DMSO or 100 nM NHWD-870 for 48 h) were seeded into the top chamber (transwell size: 0.4 μm), and TAMs (Mac, 40,000 cells per 24-well) in medium with PBS or 10 ng/ml CSF1 (**c**), or 50 ng/ml IgG or anti-CSF1R antibodies (**d**), were seeded into the bottom chamber. Shown are quantification of TAMs (**c**, **d**). Data are presented as mean ± SEM from three independent experiments. **e** Representative western blot analysis of A2780 cells stably transfected with CSF1 or empty vector (EV). Selected from three independent experiments. **f** Schematics of the animal experiments using an orthotopic mouse OC model, established by intraperitoneally injecting $1 \times 10^6$ (**g–j**) or $2 \times 10^6$ (**k–m**) human A2780 cells overexpressing CSF1 or control into female recipient nude mice. Mice were then treated with/without liposomal clodronate (LC) for 45 days (**g–j**) or treated with/without NHWD-870 (PO, QD, 3 mg/kg) for 37 days (**k–m**). **g** Increased mouse body weight was measured at the indicated time points (day 0-45 after LC treatment). **h–j** Net tumor weight (**h**), ascitic fluid volumes (**i**) tumor implantations in peritoneum, omentum and pevic cavity (**j**) were measured at day 45 from LC treatment. **k–m** Net tumor weight (**k**), ascitic fluid volumes (**l**) tumor implantations in peritoneum, omentum and pevic cavity (**m**) were measured at day 37 after NHWD-870 treatment. For **g–m**, data are presented as mean ± SEM from five different mice. $p$ values were calculated using two-tailed, unpaired $t$ tests in this figure. n.s, not significant; *$p < 0.05$; **$p < 0.01$; and ***$p < 0.001$. Source data are provided as a Source Data file.

mouse body weight (a surrogate marker of tumor burden), net tumor weight, ascitic fluid volume, and tumor implantation in peritoneum, omentum and pelvic cavity (Fig. 6g–m). LC not only significantly reduced mouse body weight, net tumor weight, ascitic fluid volume and tumor implantation in these tissues at base line, but also completely abolished the effects of CSF1 overexpression (Fig. 6g–j). On the other hand, CSF1 over-expression strongly mitigated the tumor inhibitory effects of NHWD-870 on A2780 tumors, as measured by net tumor weight, ascitic fluid volume and tumor implantation (Fig. 6k–m). Taken together, these data showed that CSF1 is sufficient to promote OC growth by increasing the proliferation of TAMs and its down-regulation in tumor cells contributes significantly to the tumor inhibitory effects of NHWD-870.

**BRD4 induces CSF1 expression through HIF1α.** Hypoxia-inducible factors (HIFs) are the master regulators of hypoxia-inducible genes, such as VEGFA and GLUT1[42,43]. Intratumor hypoxia was shown to induce macrophage recruitment through HIF-dependent CSF1 secretion by tumor cells[41]. A recent paper showed that ZMYND8/RACK7 interacts directly with HIFα and BRD4 to promote transcription elongation of some HIF target genes in breast cancer cells[44]. Thus, we asked whether BRD4 and HIF1α mediate CSF1 downregulation by NHWD-870. To this end, we first established BRD4 knockout A375 melanoma and A2780 OC cells using the CRISPR/Cas9 system and exposed these cells and the control cells to normoxia (20% $O_2$) or hypoxia (1% $O_2$) to induce HIF1α protein for 24 h (Fig. 7a, b). BRD4 deletion significantly decreased HIF1α protein under hypoxia (Fig. 7a, b), and BRD4 inhibition by NHWD-870 treatment also decreased HIF1α protein (Fig. 7c, d). To determine whether HIF1 tran-scriptional activity is affected by BRD4 deletion or inhibition, we performed the HIF1 luciferase reporter assays. BRD4 knockout, NHWD-870 or OTX-015 treatment significantly decreased HIF1 reporter activity in A2780 and A375 cells (Fig. 7e and Supple-mentary Fig. 15a). Furthermore, CSF1 mRNA levels were sig-nificantly reduced by BRD4 deletion or NHWD-870 treatment (Fig. 7f). These results raised the possibility that BRD4 regulates CSF1 expression through direct regulation of HIF1A (HIF1α gene) transcription. Consistent with this possibility, NHWD-870 treatment decreased HIF1A mRNA levels (Fig. 7g). Analysis of published ChIP-seq data[45,46] showed that BRD4 bound the HIF1α promoter in A375 cells and BRD4 inhibition by JQ1 reduced BRD4 binding to the HIF1α promoter in MDA-MB 231 cells (Fig. 7h). Importantly, ChIP-qPCR analyses showed that BRD4 bound to the HIF1A promoter and BRD4 inhibition by NHWD-870 decreased BRD4 binding to the HIF1A promoter in A2780 cells (Fig. 7i). We next asked whether HIF1α over-expression is sufficient to rescue CSF1 downregulation by

BRD4 loss. Overexpression of wild type HIF1α under hypoxic condition significantly increased mRNA levels of VEGFA, a well-known HIF target genes, as well as CSF1 in BRD4 knockout cells to the levels even much higher than those in the control cells (Fig. 7j and Supplementary Fig. 15b–e). These data indicated that BRD4 directly regulated HIF1α to induce CSF1 expression in tumor cells, which promotes CSF1R signaling in TAMs (Fig. 7k).

**p-BRD4, CSF1, and TAMs were negatively associated with patient outcome.** To investigate whether BRD4 activation in ovarian cancer cells correlates with CSF1 expression, the number of TAMs in human ovarian tumors, and patient prognosis, we examined the p-BRD4 and CSF1 levels in tumor cells and the number of TAMs using peritoneal implantation metastasis tissues from 128 human ovarian cancer (OC) patients with treatment and survival information[29]. The demographic and clinical char-acteristics of patients were described in Supplementary Table 5. Immunohistochemical staining of macrophages (CD68+ cells), p-BRD4 and CSF1 showed that macrophages were present in nearly all these implantation tumors, and p-BRD4 and CSF1 were expressed mostly in tumor cells (Supplementary Fig. 16a). Moreover, the numbers of macrophages were significantly higher in metastasis tissues than those in the primary tumors (data no shown). The number of CD68+ cells increased with lymphovas-cular invasion ($p = 0.0027$), from well to poor differentiated OC ($p < 0.0001$) (Supplementary Table 5), suggesting that poor dif-ferentiated OC attract more macrophages in implantation as compared to well-differentiated OC. However, there was no sig-nificant correlation of TAM counts with ascites ($p = 0.754$) and chemotherapy regimen ($p = 0.14$) (Supplementary Table 5). Correlation analyses showed that the CSF1 levels correlated sig-nificantly with the p-BRD4 levels in tumor cells ($r = 0.5122$) (Supplementary Fig. 16b and Supplementary Table 5). Moreover, both the p-BRD4 and CSF1 levels correlated significantly with the numbers of TAMs in human ovarian tumors ($r = 0.6976$ and $r = 0.4213$, respectively) (Supplementary Fig. 16c, d and Supple-mentary Table 5). Uni- and multi-variate analysis showed that patients with high p-BRD4 level, high CSF1 level, or high per-centage of CD68+ cells (≥11%) in tumors have significantly lower five-year overall survival rate and higher hazard ratio (Supple-mentary Tables 6, 7).

## Discussion
Most BET inhibitors in the clinic, including GSK525762, CPI-0610, OTX-015, and MT1, have similar structures as JQ1 and exhibited good preliminary effects on hematological malig-nancies, but not on most solid tumors, likely due to limited

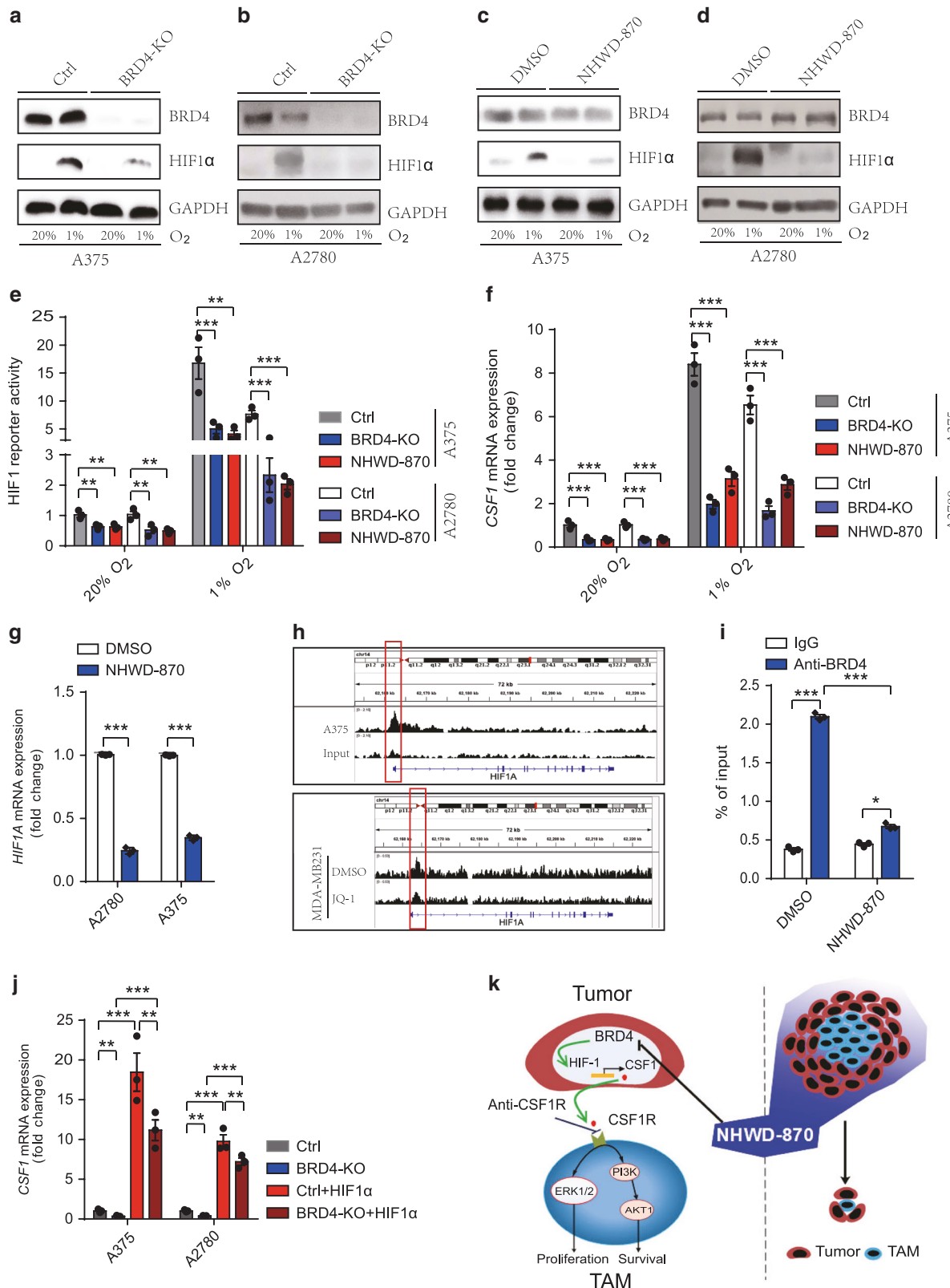

potency and drug resistance mechanisms[37,40,47–51]. However, BMS-986158, currently tested in phase I/II clinical trials, adopts a different chemotype and will likely achieve better effects. Here we developed a series of BRD4 inhibitors based on the core structure of BMS-986158. Our optimization effects led to the development of NHWD-870, which is more potent than several clinical stage BET inhibitors in cellular assays. As expected for potent BRD4

inhibitors, we observed significant inhibition of BRD4 activation and c-MYC expression by NHWD-870. In addition, due to its unique structure, NHWD-870 is orally available, exhibited acceptable metabolic stability and plasma protein binding capacity, and displayed good absorption profile of NHWD-870 in lung and tumor. These properties led to its robust single agent activity in cancer cell lines derived from solid tumors in vivo.

**Fig. 7 NHWD-870 inhibited CSF1 expression through suppressing BRD4 and HIF1α in tumor cells. a, b** Representative western blot analysis of control (Ctrl) and BRD4 knockout (KO) A375 and A2780 cells exposed to 20% or 1% $O_2$ for 24 hours. Selected from three independent experiments. **c, d** Representative western blot analysis of A375 and A2780 cells treated with DMSO or 25 nM NHWD-870 and exposed to 20% or 1% $O_2$ for 24 h. Selected from three independent experiments. **e** Relative HIF1 luciferase reporter activity in control (Ctrl), BRD4 knockout (KO), or NHWD-870 treated A375 and A2780 cells exposed to 20% or 1% $O_2$ for 24 h. Data are presented as mean ± SEM from three independent experiments. **f** RT-qPCR analyses of *CSF1* in control (Ctrl), BRD4 knockout (KO), or NHWD-870 treated A375 and A2780 cells exposed to 20% or 1% $O_2$ for 24 hours. Data are presented as mean ± SEM from three independent experiments. **g** RT-qPCR analyses of *HIF1A* in A375 or A2780 cells treated with 15 nM NHWD-870 or DMSO control for 24 h. Data are presented as mean ± SEM from three independent experiments. **h** Genome browser view of BRD4 ChIP-seq peaks on the *HIF1A* promoter. Shown are BRD4 ChIP and input control of A375 cells (top panel) and BRD4 ChIP of MDA-MB231 cells treated with DMSO or JQ-1 (bottom panel). **i** BRD4 ChIP-qPCR analyses of the *HIF1A* promoter in A2780 cells treated with DMSO or NHWD-870 for 72 h. Data are presented as mean ± SEM from three independent experiments. **j** RT-qPCR analyses of *CSF1* in A375 and A2780 cells were transiently transduced with HIF1α, then exposed to 1% $O_2$ for 24 h. Ctrl, Control. Data are presented as mean ± SEM from three independent experiments. **k** A model showing the effects of NHWD-870 on cancer cell-TAM interaction. NHWD-870 inhibits the proliferation and survival of TAMs partly by inhibiting BRD4 activity, HIF1α expression, and CSF1 secretion by tumor cells, and downregulating CSF1R mediated ERK and PI3K/AKT signaling in TAMs. *p* values were calculated using two-tailed, unpaired *t* tests in this figure. *$p < 0.05$; **$p < 0.01$; ***$p < 0.001$. Source data are provided as a Source Data file.

TAMs promote tumor growth and cancer metastasis through the secretion of growth factors and chemo-attractants[52,53]. Depletion of peritoneal macrophages or modulation of microvascular permeability to block monocyte transmigration reduced peritoneal metastasis by OC cells[54]. We recently reported that macrophages were required for formation of peritoneal TAM-OC cell spheroid, early transcoelomic metastasis and tumor growth of OC promoted[29]. We also found that *EGF* was highly expressed in TAMs, but *EGFR* was highly expressed in ID8 OC cells[29]. We further showed that the EGF/EGFR axis was critical for proliferation of tumor cells, as TAMs provided EGF to induce EGFR-positive OC cell proliferation. Consistent with our results, it was reported that EGF from macrophage promoted invasion of breast carcinoma cells[55]. Therefore, the effects of NHWD-870 on CSF1 could affect the ability of the macrophages to proliferate and to secret EGF, which is required for optimal growth of tumor cells. These results suggest that tumor cell-TAM interaction is critical for tumor growth.

We showed that BRD4 depletion or inhibition decreased HIF1α protein level and the expression of its target genes including *VEGFA* and *CSF1*. BRD4 was shown to be recruited by ZMYND8 to promote transcription elongation of some HIF target genes in breast cancer cells[44]. Furthermore, BET inhibitor JQ1 was shown to downregulate two-thirds of HIF target genes, partly due to suppressing HIF1 binding to HIF target genes such as CA9[56]. However, JQ1 did not affect the protein levels of HIF1α and HIF2α in two breast cancer cell lines[56]. The discrepancy of these results from our findings could be due to the non-specific effects of JQ1, or the different cell lines used in the previous study. Nonetheless, all these results suggest that BRD4 regulates HIF target genes through various mechanisms and BRD4 inhibition can also be used to modulate HIF activity to treat cancer. Taking together, these results suggest that BRD4 inhibition suppresses tumor growth not only through direct inhibition of cancer cell growth, but also through blockade of the interaction between cancer cells with their supporting microenvironment (Fig. 7k).

## Methods

**Compounds**. Recombinant protein and antibodies, NHWD-530, 540, 560, 830, 840, 850, 860, 870, and BMS-986158 were synthesized by Ningbo Wenda Pharma (Ninghai, Zhejiang, China). NHWD-870 was synthesized using Stille coupling (Supplementary Fig. 2) unless otherwise indicated. JQ1 was purchased from MedChem Express (Cat # HY-13030). CSF1 was purchased from R&D Systems (Cat # 216-MC). GSK525762 (Cat #. S7189), I-BET151 (Cat # S2780) and OTX-015 (Cat # S7360) were purchased from Selleck China. p-BRD4 rabbit polyclonal antibody was a gift from Dr. Cheng-Ming Chiang, University of Texas Southwestern Medical Center and other antibodies were listed in Supplementary Table 8.

**Cell culture**. All cell lines were acquired from the American Type Culture Collection (ATCC, Manassas, VA), the Specimen Resource Core of Yale SPORE in skin cancer, or colleagues. HEK293T, MCF-10A, SKBR3, UACC812, MAD-MB361, MDA-MB231, A2780, IGROV1, SKOV3, ECC-1, Ishikawa, HELA, A549, A375, ID8, B16, and B16F10 were cultured in Dulbecco's Modified Eagle Medium (DMEM); H1975, NCI-H211, JER, NCI-H526, NCI-H69 cell were cultured in RPMI-1640 Medium; JEG-3, Hep G2, U251 and SH-SY5Y were grown in Eagle's Minimum Essential Medium (EMEM); YUSOC, YUGASP, YUAME, and YUMAC were cell lines derived from tumors of patients treated at Yale University and grown in OptiMEM plus 5% FBS; and ES-2 cell line was cultured in McCoy's 5A (Modified) Medium (Thermo Fisher Scientific, USA). All these cells were supplemented with 10% fetal bovine serum (FBS; GIBCO) and 1% antibiotics (penicillin/streptomycin). SUM149 cells were grown in Ham's F-12 medium supplemented with 5% heat-inactivated fetal bovine serum (FBS) and 1% antibiotics/antimycotics (Invitrogen, Carlsbad, CA), SUM159 was maintained in Ham's F-12 medium with 5% FBS, 5 μg/ml insulin and 1 μg/ml hydrocortisone, HUVEC cells were cultured in M199 medium with 20% FBS, 20 μg/ml Endothelial Cell Growth Supplement (ECGS), and 50 μg/ml heparin. All the cells were cultured in a humidified incubator with 5% $CO_2$ at 37 °C.

**Flow cytometry**. Flow cytometry analyses were performed using antibodies listed in Supplementary Table 8. For cell surface markers, mouse TAM suspensions were stained with mouse APC-CD45, FITC-CD11b, PE-F4/80 or APC-F4/80, CD68, and PE-CD206 antibodies for 15 min on ice. For Ki67 staining, cells were fixed in 4% paraformaldehyde and stained with FITC-Ki67. Isotype matching antibody served as the negative control. Flow cytometry was performed on a FACSCalibur (BD Biosciences). Data were analyzed with BD CellQuest Pro software version 5.1.

**RT-qPCR**. Total RNA was extracted with RNeasy plus mini kit (Qiagen), and then cDNA was synthesized by iScript™ cDNA Synthesis Kit (Bio-Rad). qPCR was performed using a CFX96 Real-time PCR Instrument (BioRad). Data were collected using CFX Manager version 3.1.1517.823 and analyzed using Microsoft Excel 2010. Relative expression levels were determined by normalization to human *GAPDH* using the $\Delta\Delta C_t$ method. Primers were listed in Supplementary Table 9.

**Isolation and culture of mouse TAMs**. CD45+CD11b+F4/80+ TAMs were isolated from peritoneal metastasis of A2780 tumor-bearing mice and cultured in RPMI-1640 supplemented with 10% FBS, 1% penicillin/streptomycin, and 0–50 ng/ml CSF1 (R&D Systems) as indicated according to standard protocols[57]. In the cancer cell-macrophage co-culture experiments, CSF1 was not added. In the transwell (Corning, pore size, 0.4 μm) co-culture experiments, tumor cells were placed into the upper chamber, and TAMs were seeded into the lower chamber.

**3D Matrigel culture**. Single clone spheroids were generated by culturing A2780, ES-2, H526 and MDA-MB231 cells for 16 h as a hanging drop over a humidified plate in a $CO_2$ incubator in their corresponding complete media. Tumor cell spheroids or macrophages were embedded in Matrigel matrix (Corning Cat # 354234), treated with JQ1, NHWD-870, or DMSO control, and imaged using Zeiss Axiovert 200 fluorescence microscope (Carl Zeiss MicroImaging; Thornwood, NY), and images were captured using Openlab3 software (Improvision, Lexington, MA) after 6 days treatment. The relative spheroid sizes were measured with ImageJ version 1.52s using the 15 representative spheroids.

**Immunofluorescence study**. Immunofluorescence staining of tumor sections was performed according to the manufacturer's instructions. Slides were imaged using a Zeiss Axiovert 200 fluorescence microscope (Carl Zeiss MicroImaging; Thornwood, NY), and images were captured using Openlab3 software (Imprecision,

Lexington, MA). Cleaved caspase 3, Ki67, CSF1, and CD68 positive cells were counted with ImageJ version 1.52s[58].

**Animal experiments**. 4–6 weeks old female BALB/c nude mice were used for most experiments, with the exception of 6–8 weeks old female C57BL/6 mice were used for B16F10 experiments, and 6–8 weeks old male BALB/c nude mice and SD rats were used for pharmacokinetic experiments. All animals were purchased from Shanghai SIPPR-Bk Lab Animal Co., Ltd. All animals were housed under a regimen of 12 h light/12 h dark cycles and specific pathogen-free conditions. $5 \times 10^6$ NCI-H526, ES-2, MDA-MB231, A2780, A375, B16F10, and TMD-8 cells (0.1 ml) were injected subcutaneously into female nude mice. The melanoma patient-derived xenograft model was derived from a human melanoma sample (*BRAF*, *NARS* and *NF1* wild type, and *p53* mutant) removed from a patient in Xiangya Hospital. Surgically removed tumor fragments were implanted subcutaneously into female NSG mice. When the tumors grew to ~1000 mm³, they were harvested and cryopreserved or immediately re-implanted for expansion or experiments. Mouse body weight and tumor sizes were measured every 3 days. Once the tumors grew to 150–200 mm³, mice were treated with the compounds as described in the Figures. $1 \times 10^6$ A2780 cells (1 ml) were injected into the abdominal cavity of nude mice. Ascitic volume and tumor weight were measured by an electronic balance. All the procedures were evaluated and approved by the Institutional Animal Care and Use Committee of Central South University and Yale University.

**Statistical analysis**. Statistical analyses were performed with SAS software (version 9.1.4, SAS Institute, Cary, NC), with most data were analyzed with using two-tailed Student's *t*-tests. The correlation between CD68 positive cells, CSF1 expression, and p-BRD4 expression was assessed by the odds ratio (OR) that was estimated using logistic regression with covariate adjustment. Overall survival (OS) of patients was defined by the interval from the date of primary cytoreductive surgery to the date of death, regardless of the cause or the last follow-up appointment. The survivors were censored at the date of last contact. Logistic regression analysis and receiver operating characteristic curves were used to determine the predictive value of p-BRD4 and CD68 positive cells. The area under the curve and the 95% confidence interval were used to assess the discriminatory power of CD68 positive cells for predicting prognosis. Kaplan-Meier analysis was used to determine the OS. The log-rank test was used to compare survival outcomes between different groups of patients. p-values less than 0.05 were considered statistically significant.

**Reporting summary**. Further information on research design is available in the Nature Research Reporting Summary linked to this article.

## Data availability

The ChIP-seq data referenced in this study are available under GSE89129 and GSE102409 in Gene Expression Omnibus. All the data supporting this study are available within the article, the Supplementary file, the Source Data file, and from the corresponding authors upon reasonable request, as indicated in the Reporting Summary for this article.

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

## Acknowledgements

We thank Dr. Cheng-Ming Chiang (University of Texas Southwestern Medical Center) for generously providing p-BRD4 antibody. We thank Dr. David Stern (Yale University) for providing JQ1. We thank Dr. Ruth Halaban (Yale University) for kindly providing the human melanoma cell lines used in this study. We thank the members of the Yan laboratory at Yale University for their helpful discussions and support. The research was supported in part by Department of Defense Breast Cancer Research Program Award W81XWH-15-1-0117 (to Q. Yan), NCI P50CA121974 and R01CA237586 (to Q. Yan), The Sokoloff Family-MRA Team Science Award (to Q. Yan), and Natural Science Foundation of China General Program grant 81874138 (to M.Y.), 81728014 (to Y.T.), and Major Projects of International Cooperation and Exchanges grant 81620108024 (to X.C.).

## Author contributions

M.Y. and Q. Yan designed the project, provided study guidance, analyzed the experiments, and wrote the paper. X.C., Y.T., N.W., and C.P. helped with project design and provided guidance on some experiments. M.Y., Y.G., R.H., W.L.C., Y.L., S.P., H.S., J.L., R.Y., and Q. Yang performed the experiments, data analysis and contributed to the writing of the paper. N.W. synthesized most compounds.

## Competing interests

N.W. is the founder of Ningbo Wenda Pharma. Other authors declare no competing interests.
