## [Peer Review File · Nature Communications]

Reviewers' comments:

Reviewer #1 (Expertise: BET inhibitors, cancer, Remarks to the Author):

In this manuscript by Mingzhu Yin et al., the authors report their identification and characterization of a new BET bromodomain protein inhibitor NHWD-870, and claim that 870 possesses higher anti-tumor potency than JQ-1 and other BETi. They also claim that one major mechanism of action (MOA) of 870 is its activity to downregulate CSF1 gene expression in tumor cells which in turn leads to inhibition of TAM proliferation through CSF1 receptor signaling. Furthermore, they provide correlative data from IHC analysis of clinical specimens that tumor level of p-Brd4 is associated with the number of TAMs infiltrated in ovarian tumor peritoneal implant metastatic tissue. Overall, the data of anti-tumor potency from the selected cell culture and cell line-derived xenograft tumor models are strong and convincing. The data of PK and oral bioavailability characterization also look very good. The correlation between p-Brd4 and TAM is interesting. However, the manuscript lacks truly new mechanistic information on either the function of Brd4 or the MOA of BETi. A large number of publications have addressed the functional mechanisms of Brd4 or related BET proteins in cancer. Several BETi including ones currently on clinical trials have also been reported and many of them provided extensive information on the MOA of the BETi. This manuscript does not appear to provide data that are truly novel and significant to the field of BET or cancer epigenetics/therapeutics development. Thus, this manuscript is more appropriate for journals such as JMC or ACS series (given the strong emphasis of the presented data on the compound identification and pharmacological characterization).

Other major comments:

1). The manuscript did not present any data to demonstrate that the new compound 870 truly acts through inhibition of Brd4 in cell culture or tumor. At minimum, approaches such as gene expression profiling should be used to demonstrate how the effects of the compound and the effect of genetic silencing of Brd4 can be correlated. Data from comparison analysis with a few other BETi in this regard is highly desirable to convince readers that compound 870 is superior than the other BETi.

2). Although CSF1 is shown to be significantly affected by the compound, it lacks data to show whether CSF1 or related signaling genes is a direct target of Brd4, Myc or other related factors, and it also lacks data to show how compound 870 affects the expression of CSF1.

3). The effects on TAM in tumors were obtained with immune-deficient nude mice. Experiments for examining the effects on TAM in tumors should be performed with immune-intact mice with suitable mouse tumors. Immune deficiency can significantly skew the host immune response to the tumor growth/metastasis or the treatment.

4). Use of PDX models for some of the in vivo experiments is highly desirable, given the widely recognized value of PDX models and the limitations of cell line-based systems in cancer research.

Other comments:

a. Given that CSF1 is the major target downregulated by the compound, it will help strengthen the manuscript if analysis of CSF1 and p-Brd4 in the tumors demonstrate correlation similar to that between p-Brd4 and TAMs.

Reviewer #2 (Expertise: TME, TAMs, cancer, Remarks to the Author):

The manuscript entitled, "Potent BET inhibitor suppresses cancer cell-macrophage interaction" by M. Yin et al, describes a novel bromodomain protein (BRD4) inhibitor with improved sensitivity and reduced overall toxicity for cancer therapy. The authors convincingly demonstrate the anti-tumor and favorable pharmacodynamics of this new compound, NHWD-870. The authors demonstrate that by inhibiting proliferation and transcription in tumor cells, the drug has indirect effects on tumor-associated macrophages, thereby enhancing the anti-tumor properties of the novel drug. The assessments of the drug-like properties or the in vivo efficacies of this novel compound, including its effects on reducing the number of TAMs associated with tumors, are very well-supported by the studies presented in this paper.

However, the conclusions regarding effects of NHWD-870 on macrophages are not completely supported by the data. The authors use three non-standard approaches to study tumor associated

macrophages in this paper: 1) flow cytometric staining for CD11b+ CD206+ cells; 2) IHC of mouse macrophages for CD68 and Ki67; 3) in vitro macrophage colony forming assays.

1) The flow cytometry scheme is non-standard, as most TAMs are identified as CD11b+ F4/80+ and then classified as CD206+ or CD206-. It is incumbent on the authors to demonstrate that CD11b+CD206+ cells are identical to CD11b+F4/80+ TAMs. As the authors use CD68 in tissue staining to identify macrophages, they should also demonstrate that CD11b+CD206+ and CD11b+F4/80+ macrophages are also CD68+ by FACs and IHC.

2) CD68 is usually used as a biomarker of macrophages in human tissue, while F4/80 is usually used in mouse tissue IHC. The authors should demonstrate that CD68 is an equivalent marker to F4/80 in mouse tissues, ideally using IHC and Facs. Additionally, the high power images of tissues stained for CD68 and Ki67 in Figure 4e do not support the conclusions that NHWD-870 inhibits macrophage proliferation in vivo, as very few CD68+ cells in the field of view are Ki67+. In fact, few cells in the tissue section shown are Ki67+. The authors should show lower power fields of view and also evaluate Ki67 expression in macrophages by flow cytometry, which is easily accomplished. Additionally, the data presented in 4j-m do not convincingly show any Ki67+CD68+ cells in any condition. Therefore, the graphs showing CD68+Ki67+ cell decrease in NHWD-870 treated tissues are not supported by the data. Use of Facs analysis for percentages of Ki67+ macrophages in treated tumors would be more convincing.

3) The authors describe use of a novel 3D colony-forming assay of macrophage proliferation ex vivo but provide few if any details about this procedure in the methods section or text. Why culture these cells in 3D systems when they proliferate quite well and can be easily quantified in 2D plastic systems? What are the culture media (do they include mCSF and serum?) The authors should describe in more detail the characteristics of these isolated TAMs and the resulting colony cells. The 3D colony formation by TAMs has not been well described before. Do these cells maintain lineage (macrophage) markers? What drives their proliferation? It is likely that colony formation is dependent on Myc and possibly BRD4. In fact, the authors do show that NHWD-870 directly inhibits colony size. What about colony numbers? That is the usual metric of colony forming assays. It would be helpful also to quantify the direct effect of NHWD-870 on TAM proliferation ex vivo using a 2D growth assays in order to accurately assess the direct effects of the novel BRD4 inhibitor on macrophage proliferation. TAMs can be cultured in mCSF containing medium for at least 2 weeks on plastic dishes, so it would be best to also perform a standard quantifiable proliferation curve of TAMs ex vivo in the presence of NHWD-870. It is also possible to explore lineage marker preservation using this system ex vivo.

Additionally, for comparison purposes, it would be a good idea to compare the direct effect of NHWD-870 on in vitro bone marrow derived macrophage proliferation as well (as JQ1 is well known to inhibit bone marrow derived macrophage proliferation and gene expression in vitro).

The authors demonstrate that inhibition of tumor cell BRD4 by NHWD-870 suppresses tumor cell expression of mCSF. Using an ill-defined co-culture system, they show that NHWD-870 pre-treatment of tumor cells blocks mCSF expression and blocks proliferation of macrophages *ex vivo*. It is not possible from these studies to conclude if NHWD-870 mediated inhibition of mCSF expression is responsible for the suppression of colony formation shown in Figure 5d-e. In part, this is because the culture conditions for this unusual assay have not been described in the methods or text sections. What are the components of the culture medium? Does it include serum, mCSF and other growth factors? A more direct method of testing the effect of tumor cell expressed mCSF on TAMs *ex vivo* is to incubate TAMs in defined tumor cell conditioned media from untreated and treated tumor cells (the media placed on the macrophages should not include the drug, however). However, there is one other key weakness to these studies that affects the interpretation of these results: the authors use a concentration of NHWD-870 that kills close to 100% of tumor cells (as shown in Figure 2B). It is thus not clear if absence of mCSF or products from dying cells suppress macrophage growth as colonies *ex vivo*.

In conclusion, the authors clearly show that a novel BRD4 inhibitor is a potent anti-tumor agent but do not convincingly identify a mechanism by which it affects macrophages. Improved biomarker analysis and improved *in vitro* culture systems may resolve these issues.

Minor:

Please add exact p values to all figures or figure legends.

Please include absolute mRNA expression values for Figure 5f.

Please also include protein expression data (ELISA assay for secreted mCSF from the cell lines shown in Figure 5f).

Reviewer #3 (Expertise: Medicinal chemistry/pharmacology, Remarks to the Author):

The manuscript by Yin and co-workers describe the synthesis and characterization of the novel BRD4 inhibitor NHWD-870, and further that this compound inhibits tumor growth and proliferation of tumor-associated macrophages (TAMs).

NHWD-870 is structurally closely related to the previous compounds BMS-986158 that is currently in phase I/IIa clinical trials. Essentially, the difference that a 2-hydroxy-2-propyl substituent is replaced by a fused N-methylpyrazole. The goal of the project was to develop a compound with improved efficacy and pharmacokinetic properties relative to BMS-986158. It is unclear in which respects the properties of BMS-986158 were insufficient. Of the 8 new compounds included, NHWD-870 was 3-fold more potent than BMS-986158. NHWD-870 inhibited BRD4 phosphorylation, c-MYC expression and several cancer cell lines with higher potency than the very early BRD4 inhibitor JQ1. It appears that BMS-986158 might have been a more appropriate benchmark compound for these studies.

Information on the number of independent studies is missing for several studies. Standard deviations or similar should be stated for IC50 values.

Synthetic protocol and some characterization is found only for NHWD-870, and only ¹H NMR and low-resolution ESI MS are provided. Additional documentation of identity and purity would be preferable. The Stille-coupling used in the last step implies a risk of toxic organotin by-products that could affect the assays. Analysis for the presence of organotin trace amount would therefore be reassuring. Protocols and at least rudimentary characterization should be included also for the remaining compounds, even if they are discussed only in two sentences in the manuscript.

Of minor importance, the diagram on page 56 in the supplementary information contains the same information as Supplementary Figure 2. This can be removed. Yields should be given in the protocol. Information of who has supplied what should be provided under Materials or in the Acknowledgement, not in this diagram Scheme.

The pharmacokinetic properties of NHWD-870 were investigated in mice and rats, resulting in bioavailability of 70% and 24%, half-life of 1.7 h and 0.9 h, and a clearance of 35 and 50 mL/min/kg, respectively. Despite relatively modest values (especially the clearance approaches the hepatic blood flow), the pharmacokinetic properties of NHWD-870 are several places referred to as "excellent". Perhaps "acceptable" would be a preferable term.

The dosing information on the PK study in Methods section seems to be inconsistent with the information in Figure 3 and the text.

Figure 3 d and e shows comparable levels of NHWD-870 and BMS-986158 in tumors 21 days after oral administration of 3 mg/kg. This is unclear. What is the time scale relative to? If PK studies were performed with both NHWD-870 and BMS-986158, why is the other results shown only for NHWD-870.

A microsomal half-life of 21-39 min in several species (the curve for monkey in Suppl. Fig. 4 suggests a value closer to 10 min) also seem lower than ideal and corresponds to the high clearance. I disagree with the statement that “these results suggested that NHWD-870 has stable metabolic profile”.

Based on the hERG data, NHWD-870 is claimed to have “minimal cardiac toxicity”. An IC₅₀ of 5 uM, as found for NHWD-870, is generally considered to be a moderate value. This could be acceptable for a drug that is present in plasma only at very low concentration and for a serious indication like cancer, but it does not make a solid basis for claiming “minimal cardiac toxicity”.

To further evaluate toxicity, the effect of NHWD-870 on the viability of a selection of non-cancerous cells would be interesting.

The manuscript refers to BMS-986158 as a highly selective compound and elsewhere states “Here we report the discovery and characterization of NHWD-870, a potent bromodomain inhibitor for BET family members BRD2/3/4/T”, suggesting a less selective pan-BET inhibitor. However, only data on BRD4 is found in the manuscript. What is the selectivity of NHWD-870?

The manuscript further presents results showing that NHWD-870 not only inhibits tumor growth but also suppresses TAM proliferation, and they include data showing that BRD4 phosphorylation is positively correlated with TAM counts in human ovarian cancer patients and negatively associated with patient survival. The TAM suppression seems to be a consequence of inhibition of CSF1. I’m not an oncologist and not the most qualified person to judge these results, but they certainly seem interesting. Is the inhibition of CSF1 a direct consequence of BRD4 phosphorylation and a general property of all BRD4 inhibitors, or is this a special property of NHWD-870? To what extent does TAM suppression contribute to inhibition of tumor growth?

The presentation of new data is continued in the Discussion and not much space is dedicated to setting the overall results in perspective.

In summary, the manuscript appear to have two main messages, where one is a potent new BRD4 inhibitor and the other is the observation that the BRD4 inhibitor (and any BRD4 inhibitor?) also suppresses TAM and that this effect is likely to contribute to the anti-cancer activity of the compound. It is clear that NHWD-870 is potent, efficacious and interesting. It is also closely related to BMS-986158. Although several pieces of data are included showing that NHWD-870 is somewhat or slightly better than BMS-986158, it is not clear to me that NHWD-870 overall is significantly better. Importantly, unambiguous documentation of purity and identity should be included. The compound certainly deserves to be published, but the data presented here does in my opinion not warrant publication in Nature Communication. It is certainly possible that an expansion of the observation of TAM suppression and its link to tumor suppression of BRD4 inhibitors would be suitable for Nature Communication.

Yale University

Mailing address:
Department of Pathology
Yale School of Medicine
P.O. Box 208023
New Haven, CT 06520-8023
TEL: (203) 785-6672
FAX: (203) 785-2443

Shipping address:
Department of Pathology
Yale School of Medicine
310 Cedar Street, BML 348C
New Haven, CT 06510-3128

Email: qin.yan@yale.edu

Jan 2, 2019

Dear reviewers,

Thank you for carefully reviewing our manuscript NCOMMS-18-17300 entitled “Potent BET inhibitor suppresses cancer cell-macrophage interaction”. You found our initial submission interesting and important, and provided a number of constructive and insightful comments to improve our manuscript.

To address your comments, we have conducted extensive experiments that are presented in **Figures 1g, 3d-f, 5g, 6a-h, S2b, S4a-c, S10 a,b, g-i, S13 a, b, d, Tables S5-7** and **Reviewer Figures 1-4** described below. To summarize, the main experiments included in the revised manuscript are: (1) Comparison of NHWD-870 with three other clinical stage BETi in cellular assay (**Fig 1g**); (2) Efficacy data for NHWD-870 in immunocompetent mouse melanoma model, human melanoma model (compared against BMS-986158) and melanoma PDX model (**Fig 3d-f**); (3) new regulatory mechanism of CSF1 by BRD4 through HIF1 α (**Fig 6**); (4) FACS analysis that confirmed our IF studies, based on the suggestion of reviewer 2 (**Fig S10a,b**); (5) Revised synthesis strategy based on the suggestion of reviewer 3 (**Fig S2b**); (6) Analysis of CSF1 in human patient cohorts (**Fig S13a, b, d, Tables S5-7**). These new data directly addressed the major concerns by adding additional novel biological mechanistic insights and providing new data to substantiate our conclusions. Detailed information can be found in the response to all three reviewers.

We also modified the text and added additional method and discussion sections according to the reviewers’ suggestions and critiques. To facilitate your review of the revised manuscript, we marked the major change of the manuscript in **red**. Please see below for our point-by-point response to all reviewers’ comments in *italics* below. Again, thank you for your time, effort and very helpful comments, which have helped us to improve our paper.

Reviewer #1 (Expertise: BET inhibitors, cancer):

*1. Overall, the data of anti-tumor potency from the selected cell culture and cell line-derived xenograft tumor models are **strong and convincing**. The data of PK and oral bioavailability characterization also **look very good**. The correlation between p-Brd4 and TAM is **interesting**.*

We thank the reviewer for the encouraging comments.

2. However, the manuscript lacks truly new mechanistic information on either the function of Brd4 or the MOA of BETi. A large number of publications have addressed the functional mechanisms of Brd4 or related BET proteins in cancer. Several BETi including ones currently on clinical trials have also been reported and many of them provided extensive information on the MOA of the BETi. This manuscript does not appear to provide data that are truly novel and significant to the field of BET or cancer epigenetics/therapeutics development.

Thanks for your comments. We agree with the reviewer that previous reports have shown extensive MOA for BETi. However, our manuscript provides different MOA of BETi on tumor microenvironment. In our initial submission, we reported for the first time in **Fig 5** that BET inhibition suppresses proliferation of tumor associate macrophage through downregulation of CSF1 in tumor cells. In our revised manuscript, we reported in the new **Fig 6** that mechanistically, BETi or BRD4 deletion decreased HIF1 α expression to suppress CSF1 expression. These results are novel as we revealed new functions of BETi on tumor microenvironment that support tumor growth.

Fig. 6 NHWD-870 inhibited CSF1 expression through suppressing BRD4 and HIF1 α in tumor cells.

(a) Western blot analysis of control (Ctrl) and BRD4 knockout (KO) A375 cells exposed to 20% or 1% O₂ for 24 hours. (b) Western blot analysis of A375 cells treated with DMSO or 25 nM NHWD-870 and 20% or 1% O₂ for 24 hours. (c) Relative HIF1 luciferase reporter activity in control (Ctrl), BRD4 knockout (KO), or NHWD-870 treated HeLa cells exposed to 20% or 1% O₂ for 24 hours. Data were represented as mean \pm SEM (n=3). **, p<0.01; ***, p<0.001. (d,e) RT-qPCR analysis of *VEGFA* (d) and *CSF1* (e) in control (Ctrl), BRD4 knockout (KO), or NHWD-870 treated HeLa cells exposed to 20% or 1% O₂ for 24 hours. Data were represented as mean \pm SEM (n=3). ***, p<0.001. (f) Western blot analysis of control and BRD4 knockout (KO) HeLa cells transfected with HIF1 α plasmids and exposed to 1% O₂ for 24 hours. (g,h) RT-qPCR analysis of *VEGFA* (g) and *CSF1* (h) in cells shown in panel f. Ctrl, Control. ***, p<0.001. (i) A model showing the effects of NHWD-870 on cancer cell-TAM interaction. NHWD-870 inhibits the proliferation and survival of TAMs by inhibiting BRD4 activity, HIF1 α expression, and CSF1 secretion by tumor cells, and downregulating CSF1R mediated ERK and AKT signaling in TAMs.

3. *The manuscript did not present any data to demonstrate that the new compound 870 truly acts through inhibition of Brd4 in cell culture or tumor. At minimum, approaches such as gene expression profiling should be used to demonstrate how the effects of the compound and the effect of genetic silencing of Brd4 can be correlated.*

Thanks for your helpful suggestions. We have conducted additional experiments to show BRD4 deletion has similar effects as NHWD-870. In the new **Fig S4**, we showed that BRD4 deletion reduced c-MYC and decreased growth of A375 melanoma cells, similar to the effects of NHWD-870 on multiple cell lines. In the new **Fig 6**, we showed that BRD4 deletion had similar effects as NHWD870 on HIF1 α expression, HIF1 reporter activity, *VEGFA* and *CSF1* expression.

Supplementary information, Figure S4 BRD4 loss inhibited c-MYC expression, and suppressed the growth of A375 melanoma cells.

(a) Western blot analysis of control (Ctrl) or BRD4 knockout (KO1 and KO2) A375 cells with the indicated antibodies. Polyclonal cells with the corresponding sgRNAs were used in these experiments. (b,c) Clonogenic assays of control (Ctrl) or BRD4 knockout (KO1 and KO2) A375 cell. Shown are representative images (b) and quantification of colony numbers (c).

4. Data from comparison analysis with a few other BETi in this regard is highly desirable to convince readers that compound 870 is superior than the other BETi.

Thanks for your suggestion. In addition to our previous comparison with JQ1 and a clinical stage compound BMS-986158 in Fig 1e, f, we also added new Fig 1g to show that NHWD-870 is superior than three other potent BETi GSK-525762, OTX-015, and I-BET151, with the first two are clinical stage compounds.

Fig.1 Rational design of a potent BRD4 inhibitor NHWD-870. (e,f) Cellular activities of NHWD-840, NHWD-850, NHWD-860, NHWD-870, JQ-1 and BMS-986158 in SCLCs (H211) (e) and TNBCs

(MDA-MB231) (f), as measured by alamarBlue assays. Bottom panels show the average IC50 values for these compounds. (g) Cellular activities of GSK-525762, I-BET151, OTX-015 and NHWD-870 in melanoma cells (A375), as measured by alamarBlue assays. Bottom panels show the average IC50 values for these compounds. Three independent experiments were performed for all experiments. Data are plotted as mean \pm SEM (n=3).

5. Although CSF1 is shown to be significantly affected by the compound, it lacks data to show whether CSF1 or related signaling genes is a direct target of Brd4, Myc or other related factors, and it also lacks data to show how compound 870 affects the expression of CSF1.

In our revised manuscript, we show in the new **Fig 6** that BRD4 regulates CSF1 expression indirectly through HIF1 α . BETi or BRD4 deletion decreased HIF1 α expression to suppress HIF target genes including CSF1. Please also refer to our response to your comment #2.

6. The effects on TAM in tumors were obtained with immune-deficient nude mice. Experiments for examining the effects on TAM in tumors should be performed with immune-intact mice with suitable mouse tumors. Immune deficiency can significantly skew the host immune response to the tumor growth/metastasis or the treatment.

Thanks for your suggestion. In the new **Fig 3d**, we showed that NHWD-870 treatment decreased the growth of B16F10 melanoma in immunocompetent B16F10 model.

Fig.3 Oral administration of NHWD-870 strongly suppressed the growth of established lung tumor, ovarian tumor, lymphoma and melanoma in vivo.

(d) Tumor growth curves for B16F10 melanoma (n=5) bearing mice treated with the indicated compounds once daily for 11 days.

In the new **Fig s10 a,b**, we showed that NHWD-870 significantly decreased the number of TAMs in these mice using the improved FACS analysis suggested by Reviewer #2.

Supplementary information, Figure S10 NHWD-870 treatment decreased the number of TAMs in tumor models.

(a,b) Tumors from A375 tumor bearing mice in **Figure 3e** were harvested at day 21 and stained with anti-CD45, CD11b, F4/80, CD68 and Ki67 followed by FACS analysis. Shown are representative FACS plots (a) and quantification of the percentage of CD11b⁺F4/80⁺ and CD11b⁺F4/80⁺CD68⁺Ki67⁺ cells (b).

7. Use of PDX models for some of the *in vivo* experiments is highly desirable, given the widely recognized value of PDX models and the limitations of cell line-based systems in cancer research.

We agree with reviewer. We examined the effect of NHWD-870 on a melanoma PDX model and showed in the new **Fig 3f** that NHWD-870 suppresses the growth of PDX.

Fig.3 Oral administration of NHWD-870 strongly suppressed the growth of established lung tumor, ovarian tumor, lymphoma and melanoma *in vivo*.

(f) Tumor growth curves for a patient derived xenograft (PDX) of melanoma (n=5) bearing mice treated with the indicated compounds once daily (five days on, two days off) for 21 days.

8. Given that CSF1 is the major target downregulated by the compound, it will help strengthen the

manuscript if analysis of CSF1 and p-Brd4 in the tumors demonstrate correlation similar to that between p-Brd4 and TAMs.

Thanks a lot for your suggestion. These data have been added as suggested in the new **Fig S13** and **Tables S5-7**, which showed that CSF1 expression correlates with p-BRD4 and TAMs.

Supplementary information, Figure S13 The levels of phosphorylated BRD4 and CSF1 in tumor cells, as well as the number of TAMs were positively correlated in human ovarian tumors.

(a) Representative immunohistochemical staining of CD68, p-BRD4 and CSF1 in epithelial OC implantation samples with low (left panel), medium (middle panel) and high (right panel) levels of CD68, p-BRD4 and CSF1. Nuclei were stained with hematoxylin. Bar: 50 μ m. (b-d) Scatter plots showing the relationship between p-BRD4 and CSF1 intensity (b), between p-BRD4 intensity and percentage of CD68 positive cells (c), between CSF1 intensity and percentage of CD68 positive cells (d) in 128 epithelial OC implantation samples.

Reviewer #2 (Expertise: TME, TAMs, cancer):

1. The manuscript entitled, "Potent BET inhibitor suppresses cancer cell-macrophage interaction" by M. Yin et al, describes a novel bromodomain protein (BRD4) inhibitor with improved sensitivity and reduced overall toxicity for cancer therapy. The authors **convincing** demonstrate the anti-tumor and favorable pharmaco-dynamics of this new compound, NHWD-870. The authors demonstrate that by inhibiting proliferation and transcription in tumor cells, the drug has indirect effects on tumor-associated macrophages, thereby enhancing the anti-tumor properties of the novel drug. The assessments of the drug-like properties or the in vivo efficacies of this novel compound, including its effects on reducing the number of TAMs associated with tumors, are **very well-supported** by the studies presented in this paper.

We thank the reviewer for your enthusiastic comments.

2. The flow cytometry scheme is non-standard, as most TAMs are identified as CD11b+ F4/80+ and then classified as CD206+ or CD206-. It is incumbent on the authors to demonstrate that CD11b+CD206+ cells are identical to CD11b+F4/80+ TAMs. As the authors use CD68 in tissue staining to identify macrophages, they should also demonstrate that CD11b+CD206+ and CD11b+F4/80+ macrophages are also CD68+ by FACS and IHC. CD68 is usually used as a biomarker of macrophages in human tissue, while F4/80 is usually used in mouse tissue IHC. The authors should demonstrate that CD68 is an equivalent marker to F4/80 in mouse tissues, ideally using IHC and Facs.

We thank the reviewer for your suggestion. In our tumor models at the end stage, most TAMs are M2-like macrophages that are CD206+ and we isolate TAMs using CD45+CD11b+F4/80+CD206+ from tumor bearing mice. Using FACS analysis, we showed that CD11b+ TAMs are mostly F4/80+ cells (see correlation in new **Fig S10a** middle panel), CD68+ cells (see **Fig S10a** right panel), and CD206+ cells (see correlation in **Fig S10c**).

Supplementary information, Figure S10 NHWD-870 treatment decreased the number of TAMs in tumor models.

(a,b) Tumors from A375 tumor bearing mice in **Figure 3e** were harvested at day 21 and stained with anti-CD45, CD11b, F4/80, CD68 and Ki67 followed by FACS analysis. Shown are representative FACS plots (a) and quantification of the percentage of CD11b⁺F4/80⁺ and CD11b⁺F4/80⁺CD68⁺Ki67⁺ cells (b). (c,d) Tumors from mice described in **Figure 4e** were harvested at day 45 and stained with anti-CD45, CD11b and CD206 followed by FACS analysis. Shown are representative FACS plots (c) and quantification of the percentage of CD11b⁺CD206⁺ cells (d).

In our previous publication (Yin et al. JCI 2016; 126(11):4157-4173), we showed that CD68 is a reliable biomarker for myeloid cells marked by GFP (**Reviewer Fig 1** from our previous paper).

Reviewer Fig 1. CD68 marks most of the macrophages. ID8 Ovarian cancer cells stably expressing mCherry fluorescence protein were implanted into 8-week-old tomato^{LysM-Cre} recipient mice (*LysM-Cre* mice crossed to the tomato-EGFP reporter *mT/mG* mice), which labels myeloid cells by GFP. Figure showed that spheroids collected at week 8 were subjected to immunostaining with APC-conjugated (647 nm) anti-CD68 and DAPI, followed by confocal imaging. GFP⁺ and CD68⁺ macrophages, Cherry⁺ tumor cells, and DAPI for nuclear staining are shown.

In addition, another group (Gil-Bernabé' et al Blood. 2012;119(13):3164-75.) used CX3CR1-GFP to visualize monocytes/macrophages recruited to tumors. They showed in their Fig 3B and Fig 6C that CD68, F4/80, and CD11b are very similar macrophage markers in B16F10 tumor model that we used in

this paper. Taken together, multiple lines of evidence show that CD68, F4/80, CD11b and CD206 are very similar markers in our system for TAMs in late stage tumors.

3. Additionally, the high power images of tissues stained for CD68 and Ki67 in Figure 4e do not support that the conclusions that NHWD-870 inhibits macrophage proliferation in vivo, as very few CD68+ cells in the field of view are Ki67+. In fact, few cells in the tissue section shown are Ki67+. The authors should show lower power fields of view and also evaluate Ki67 expression in macrophages by flow cytometry, which is easily accomplished. Additionally, the data presented in 4j-m do not convincingly show any Ki67+CD68+ cells in any condition. Therefore, the graphs showing CD68+Ki67+ cell decrease in NHWD-870 treated tissues are not supported by the data. Use of FACS analysis for percentages of Ki67+ macrophages in treated tumors would be more convincing.

We apologize that the number of cells is few in the high power images. As suggested by the reviewer, we have now added the lower power images in the new **Fig S9e**.

To further validate our results, we have conducted FACS analysis as suggested by the reviewer in the new **Fig S10a,b**, which showed that that NHWD-870 treatment significantly decreased Ki67+ CD68+ macrophages (See Figure in Reply to Comment #2). These results are consistent with the IHC results.

4. The authors describe use of a novel 3D colony-forming assay of macrophage proliferation ex vivo but provide few if any details about this procedure in the methods section or text. Why culture these cells in 3D systems when they proliferate quite well and can be easily quantified in 2D plastic systems? What are the culture media (do they include mCSF and serum?) The authors should describe in more detail the characteristics of these isolated TAMs and the resulting colony cells. The 3D colony formation by TAMs has not been well described before. Do these cells maintain lineage (macrophage) markers?

Thanks for pointing out that our 3D-colony formation assay of macrophages is novel. We apologize for not including sufficient information in experimental procedures on macrophage isolation and culture, and have added that to the revised manuscript. The major reason why we used 3D culture is that TAMs tend to cluster together in our ovarian cancer model (Yin et al. JCI 2016; 126(11):4157-4173), therefore we believe that our 3D culture condition better mimics the in vivo condition. The main difference of this 3D culture from 2D culture is Matrigel, and we do not expect changes of macrophage lineage. We showed in our previous publication on cancer cell and macrophage 3D co-culture experiments (Yin et al. JCI 2016; 126(11):4157-4173 and Long L, Bio Protoc 2018 doi: 10.21769/BioProtoc.2815.2018), the macrophages maintained their macrophage markers in 3D-culture condition. Serum was added to the culture media, but CSF1 was only added where indicated and was not added for cancer cell-macrophage co-culture experiments, which were used to assess the effects of cytokines secreted from tumor cells on macrophage proliferation.

We agree that TAM proliferation is more easily quantifiable in 2D system and we have added in **Fig S10g-i** and assessed the effects of NHWD-870 on A2780 cancer cell-supported growth of TAMs. We showed that 50 nM NHWD-870 decreased the growth of TAMs by about 6 fold in the 2D system, which is consistent with the 3-D culture data in **Fig 5d-e**, in which 100 nM NHWD-870 decreased the growth of TAMs by about 8 fold.

Supplementary information, Figure S10 NHWD-870 treatment decreased the number of TAMs in tumor models.

(g-i) NHWD-870 significantly inhibited tumor cell supported TAM proliferation. A2780 cells (pre-treated with DMSO, 25 or 50 nM NHWD-870 for 48h) were seeded into the top chamber (transwell size: 0.4 μm) and TAMs (1×10^5 cells per 6-well) in medium were seeded into the bottom chamber. $\text{CD45}^+\text{F4}/80^+\text{CD11b}^+\text{CD206}^+$ TAMs were isolated from tumors of ovarian cancer-bearing donor mice. Shown are schematics of the experiment **(g)**, representative images **(h)** and quantification of colonies **(i)**. Data are presented as mean \pm SEM. $n=3$. ***, $p<0.001$.

5. What drives their proliferation? It is likely that colony formation is dependent on Myc and possibly BRD4. In fact, the authors do show that NHWD-870 directly inhibits colony size. What about colony numbers? That is the usual metric of colony forming assays. It would be helpful also to quantify the direct effect of NHWD-870 on TAM proliferation ex vivo using a 2D growth assays in order to accurately assess the direct effects of the novel BRD4 inhibitor on macrophage proliferation. TAMs can be cultured in mCSF containing medium for at least 2 weeks on plastic dishes, so it would be best to also perform a standard quantifiable proliferation curve of TAMs ex vivo in the presence of NHWD-870. It is also possible to explore lineage marker preservation using this system ex vivo.

This is a great question. The proliferation of these macrophages is mainly driven by CSF1-CSF1R. In Fig **S10e** NHWD870 directly inhibited colony size of macrophages by < 2 fold, which is much less dramatic than the indirect effects through suppression of CSF1 in tumor cells (~ 15 fold reduction) in Fig **5c-e**. The in vivo effects of NHW-870 are consistent with the effects through CSF1 inhibition (~ 16 fold reduction).

In **Reviewer Fig 2**, we showed that direct treatment of TAMs in 2D growth assays with NHWD-870 can only decrease CSF1 induced growth of macrophages by < 2 fold and can not override the effects of CSF1 treatment, which induces TAM proliferation by > 10 fold. Moreover, even BRD4 knockout and c-MYC knockout only decreased CSF1 induced growth of macrophages by < 3 fold. These results suggest that macrophage growth is mainly suppressed by decreasing CSF1 expression in tumor cells.

Reviewer Fig 2. NHWD-870, BRD4 deletion, and c-MYC deletion partially inhibited proliferation of TAMs induced by CSF1. (a-c) CD11b⁺F4/80⁺ macrophages in the orthotopic B16F10 melanoma model were harvested from tumor tissues. Shown are representative CD45⁺, CD11b⁺, F4/80⁺ FACS plots (a) and quantification of the percentage of Ki67⁺ cells after treatment with DMSO or NHWD-870 dependent on CSF1 stimulation (b,c). (d-g) FACS plots and quantification of the percentage of Ki67⁺ cells after CRISPR/Cas9-mediated BRD4 knockout (d,e) and c-MYC knockout (f,g) in CD11b⁺F4/80⁺ macrophages. CSF1 is added at the indicated concentration.

Please also see response to comment # 4 about our 2D assays, in which we used colony number to quantify the number of TAMs. In the 3D assays, the number of colonies did not change much after NHWD-870 treatment and the main differences are colony size.

6. For comparison purposes, it would be a good idea to compare the direct effect of NHWD-870 on *in vitro* bone marrow derived macrophage proliferation as well (as JQ1 is well known to inhibit bone marrow derived macrophage proliferation and gene expression *in vitro*).

Thanks for your suggestion. Although we did not conduct the *in vitro* experiments, we examined the effects of NHWD-870 on monocytes in peripheral blood and found that NHWD-870 treatment had relatively mild effects on these cells (Reviewer Fig 3).

Reviewer Fig 3. *NHWD-870 mildly decreased monocytes from circulation in mice.* B16F10 melanoma tumor bearing mice (n=5) treated with the indicated compounds for 11 days. Monocytes in blood of DMSO and NHWD-870-treated mice were analyzed by complete blood cell counting (CBC).

7. *The authors demonstrate that inhibition of tumor cell BRD4 by NHWD-870 suppresses tumor cell expression of mCSF. Using an ill-defined co-culture system, they show that NHWD-870 pre-treatment of tumor cells blocks mCSF expression and blocks proliferation of macrophages ex vivo. It is not possible from these studies to conclude if NHWD-870 mediated inhibition of mCSF expression is responsible for the suppression of colony formation shown in Figure 5d-e. In part, this is because the culture conditions for this unusual assay have not been described in the methods or text sections. What are the components of the culture medium? Does it include serum, mCSF and other growth factors? A more direct method of testing the effect of tumor cell expressed mCSF on TAMs ex vivo is to incubate TAMs in defined tumor cell conditioned media from untreated and treated tumor cells (the media placed on the macrophages should not include the drug, however). However, there is one other key weakness to these studies that affects the interpretation of these results: the authors use a concentration of NHWD-870 that kills close to 100% of tumor cells (as shown in Figure 2B). It is thus not clear if absence of mCSF or products from dying cells suppress macrophage growth as colonies ex vivo.*

Thanks for your suggestion and we again apologize for omitting information about the experiments, which has been added and addressed in response to comment #4. We agree that the experiments can be done with conditioned media. The importance of CSF1 was shown by our CSF1 rescue experiment in **Fig 5m-n** and CSF1R blockade experiment in **Fig 5o**. These results suggest that the decreased macrophage growth after treatment of tumor cells with NHWD-870 is not simply due to dying cells.

Fig.5 NHWD-870 downregulated CSF1 expression in tumor cells to inhibit TAM proliferation through CSF1R signaling.

(m-o) A2780 cells (pre-treated with DMSO or 100 nM NHWD-870 for 48h) were seeded into the top chamber (transwell size: 0.4 μ m) and TAMs (Mac, 40,000 cells per 24-well) in medium with PBS or 10 ng/ml CSF1 (n), 50 ng/ml IgG or anti-CSF1R antibodies (o), were seeded into the bottom chamber. Shown are schematics of the experiments (m) and quantification of TAMs (n, o). Data are presented as mean \pm SEM. n=3. ***, p<0.001.

We apologize for misleading labeling for the 5-day MTT assays in **Fig 2b** as Y axis should be “relative cell number”. In these experiments, 100 nM NHWD-870 strongly suppressed the growth of these cells after 5 days. However, treatment of these cells with 100 nM NHWD-870 did not cause cell death after 48-hour treatment. In some of our new experiments, we decreased the dosage of NHWD-870 to 25 nM and 50 nM and obtained consistent results.

8. In conclusion, the authors **clearly** show that a novel BRD4 inhibitor is a potent anti-tumor agent but do not convincingly identify a mechanism by which it affects macrophages. Improved biomarker analysis and improved in vitro culture systems may resolve these issues.

Thanks for your encouraging comments. We hope that we have addressed your concerns on the biomarker and in vitro culture assays. In addition, we added additional mechanistic insights in the new **Fig 6** (see reply to Reviewer 1 Comment #2).

9. Minor: Please add exact p values to all figures or figure legends.

Thanks for your suggestion. However, we respectfully submit that classification of p values in different categories is easier for the readers and was used in most research articles.

10. Please include absolute mRNA expression values for Figure 5f.

Thanks for your suggestion. We have added the relative mRNA expression values in the new **Fig 5f**, which allows comparison of mRNA levels across the cell lines.

11. Please also include protein expression data (ELISA assay for secreted mCSF from the cell lines shown in Figure 5f).

Thanks for your suggestion. We have added ELISA data in the new **Fig 5g**. The results are quite consistent with the mRNA level in **Fig 5f**.

Fig.5 NHWD-870 downregulated CSF1 expression in tumor cells to inhibit TAM proliferation through CSF1R signaling.

(f) RT-qPCR analysis of relative *CSF1* mRNA level in ovarian cancer cells (ID8, A2780, SKOV3 and ES-2) and melanoma cells (B16, YUSOC, YUGASP, YUAME, YUMAC and A375) treated with 50 nM NHWD-870 for 48 h. Data are presented as mean \pm SEM. n=3. ***, p<0.001. (g) CSF1 protein levels in supernatant of 10⁶ ID8, B16, A2780, SKOV3 and A375 cells treated with DMSO or 25 nM NHWD-870 for 24h, as measured by ELISA. Data are presented as mean \pm SEM. n=3. **, p<0.01; ***, p<0.001.

Reviewer #3 (Expertise: Medicinal chemistry/pharmacology):

1. The manuscript by Yin and co-workers describe the synthesis and characterization of the novel BRD4 inhibitor NHWD-870, and further that this compound inhibits tumor growth and proliferation of tumor-associated macrophages (TAMs). NHWD-870 is structurally closely related to the previous compounds BMS-986158 that is currently in phase I/IIa clinical trials. Essentially, the difference that a 2-hydroxy-2-propyl substituent is replaced by a fused N-methylpyrazole. The goal of the project was to develop a compound with improved efficacy and pharmacokinetic properties relative to BMS-986158. It is unclear in which respects the properties of BMS-986158 were insufficient. Of the 8 new compounds included, NHWD-870 was 3-fold more potent than BMS-986158. NHWD-870 inhibited BRD4 phosphorylation, c-MYC expression and several cancer cell lines with higher potency than the very early BRD4 inhibitor JQ1. It appears that BMS-986158 might have been a more appropriate benchmark compound for these studies.

We agree with the reviewer about the assessment that BMS-986158 is a good benchmark compound. However, this paper is the first publication of both BMS-986158 and NHWD-870, but JQ1 has been well characterized for many previous mechanistic studies. Therefore, we compare NHWD-870 against JQ1 in some mechanistic studies to confirm its mechanism of action. We did compare NHWD-870 with BMS-986158 in Fig 1e, f, 3a, S6c,d, 7c, Table S4, and the new Fig 3e and showed NHWD-870 has better effects and less toxicity.

Fig.3 Oral administration of NHWD-870 strongly suppressed the growth of established lung tumor, ovarian tumor, lymphoma and melanoma in vivo.

(e) Tumor growth curves for A375 melanoma (n=6) bearing mice treated with the indicated compounds once daily (five days on, two days off) for 21 days. Data are presented as mean \pm SEM in all figures. **, p<0.01;

In addition, we also added new **Fig 1g** to show that NHWD-870 is superior than three other potent BETi GSK-525762, OTX-015, and I-BET151, with the first two are clinical stage compounds.

Fig.1 Rational design of a potent BET inhibitor NHWD-870.

(e,f) Cellular activities of NHWD-840, NHWD-850, NHWD-860, NHWD-870, JQ-1 and BMS-986158 in SCLCs (H211) (e) and TNBCs (MDA-MB231) (f), as measured by alamarBlue assays. Bottom panels show the average IC₅₀ values for these compounds. (g) Cellular activities of GSK-525762, I-BET151, OTX-015 and NHWD-870 in melanoma cells (A375), as measured by alamarBlue assays. Bottom panels show the average IC₅₀ values for these compounds. Three independent experiments were performed for all experiments. Data are plotted as mean ± SEM (n=3).

2. Information on the number of independent studies is missing for several studies. Standard deviations or similar should be stated for IC₅₀ values.

Thanks for pointing that out. All experiments in Fig 1 were performed three times as indicated in the Figure legends. IC₅₀s±SEM have been added to the revised figures.

3. Synthetic protocol and some characterization is found only for NHWD-870, and only ¹H NMR and low-resolution ESI MS are provided. Additional documentation of identity and purity would be preferable. The Stille-coupling used in the last step implies a risk of toxic organotin by-products that could affect the assays. Analysis for the presence of organotin trace amount would therefore be reassuring. Protocols and at least rudimentary characterization should be included also for the remaining compounds, even if they are discussed only in two sentences in the manuscript. Of minor importance, the diagram on page 56 in the supplementary information contains the same information as Supplementary Figure 2. This can be removed. Yields should be given in the protocol. Information of who has supplied what should be provided under Materials or in the Acknowledgement, not in this

diagram Scheme.

Thanks for your suggestion. We have now included synthesis pathway for all the NHWD compounds in the supplementary information. The suggestion about staying away from Stille-coupling is a very good suggestion. As suggested by the reviewer, when we were planning for clinical studies, we have revised this step of synthesis to Suzuki coupling (see **Fig S2b**). We have also added another step to add HCl to the compound to facilitate absorption. Duplicate information has been removed and yields were added in this diagram.

Supplementary information, Figure S2 Schematic representation of chemical synthesis of NHWD-870 and NHWD-870-HCl.

Shown are the original synthesis procedure (a) and the revised synthesis procedure (b).

We have used these NHWD-870-HCl compounds in the new **Fig 3d-f** and observed similar results. Please see “supplementary file for reviewer 3.pdf” for detailed validation of NHWD-870-HCl compound (also called C17033155-G). We have also included yield in the new synthesis and revised the method section as suggested by the reviewer.

Fig.3 Oral administration of NHWD-870 strongly suppressed the growth of established lung tumor, ovarian tumor, lymphoma and melanoma in vivo.

(d) Tumor growth curves for B16F10 melanoma (n=5) (d) bearing mice treated with the indicated compounds once daily for 11 days. (e,f) Tumor growth curves for A375 melanoma (n=6) (e), and patient derived xenograft (PDX) of melanoma (n=5) (f) bearing mice treated with the indicated compounds once daily (five days on, two days off) for 21 days. For panels d-f, NHWD-870 stands for NHWD-870-HCl. Data are presented as mean \pm SEM in all figures. *, p<0.05; **, p<0.01; ***, p<0.001.

4. The pharmacokinetic properties of NHWD-870 were investigated in mice and rats, resulting in bioavailability of 70% and 24%, half-life of 1.7 h and 0.9 h, an a clearance of 35 and 50 mL/min/kg, respectively. Despite relatively modest values (especially the clearance approaches the hepatic blood flow), the pharmacokinetic properties of NHWD-870 are several places referred to as "excellent". Perhaps "acceptable" would be a preferable term.

Thank you for your suggestion. We have revised our statement to "acceptable" as suggested by the reviewer.

5. The dosing information on the PK study in Methods section seems to be inconsistent with the information in Figure 3 and the text.

Thanks for pointing out the inconsistency. The information in the Figure and text is correct. We have corrected the information in the revised methods.

4. Figure 3 d and e shows comparable levels of NHWD-870 and BMS-986158 in tumors 21 days after oral administration of 3 mg/kg. This is unclear. What is the time scale relative to? If PK studies were performed with both NHWD-870 and BMS-986158, why is the other results shown only for NHWD-870.

Sorry for not explaining the experiments well. The time scale the time after injection on day 21. Mice were orally administrated the compounds every day for 21 days. We have now revised the text to make it more clear. As this paper is focused on NHWD-870, we only compared NHWD-870 and BMS-986158 in certain key experiments.

5. A microsomal half-life of 21-39 min in several species (the curve for monkey in Suppl. Fig. 4 suggests a value closer to 10 min) also seem lower than ideal and corresponds to the high clearance. I disagree with the statement that “these results suggested that NHWD-870 has stable metabolic profile”.

We agree with the reviewer that we will need to be conservative about our interpretation and have revised our statement to “acceptable metabolic profile”.

6. Based on the hERG data, NHWD-870 is claimed to have “minimal cardiac toxicity”. An IC₅₀ of 5 μ M, as found for NHWD-870, is generally considered to be a moderate value. This could be acceptable for a drug that is present in plasma only at very low concentration and for a serious indication like cancer, but it does not make a solid basis for claiming “minimal cardiac toxicity”.

We agree with the review that this should be considered a moderate value. As suggested by the reviewer, we have modified it to “tolerable toxicity”.

7. To further evaluate toxicity, the effect of NHWD-870 on the viability of a selection of non-cancerous cells would be interesting.

This is an important question. We have assessed the effects of NHWD-870 on non-cancerous immortalized keratinocytes and found even 1 μ M NHWD-870 had minimal effects of the viability of these cells.

Reviewer Fig 4. NHWD-870 does not suppress the growth of non-cancerous cells. MTT assays of HaCAT cells (Human immortalized keratinocytes) treated with 0 – 1,000 nM NHWD-870 as indicated for 24h, 48h or 72h.

8. The manuscript refers to BMS-986158 as a highly selective compound and elsewhere states “Here we report the discovery and characterization of NHWD-870, a potent bromodomain inhibitor for BET family members BRD2/3/4/T”, suggesting a less selective pan-BET inhibitor. However, only data on BRD4 is found in the manuscript. What is the selectivity of NHWD-870?

Thanks for pointing this out. We assumed that the specificity of NHWD-870 is similar to that of BMS-986158, a BRD2/3/4/T inhibitor as described in the BMS patent. However, we only conducted BRD4 binding assays and should not make that claim. We have revised our manuscript to state that NHWD-970 is a BRD4 inhibitor, and its specificity against other bromodomain proteins remain to be determined.

*9. The manuscript further presents results showing that NHWD-870 not only inhibits tumor growth but also suppresses TAM proliferation, and they include data showing that BRD4 phosphorylation is positively correlated with TAM counts in human ovarian cancer patients and negatively associated with patient survival. The TAM suppression seems to be a consequence of inhibition of CSF1. I’m not an oncologist and not the most qualified person to judge these results, but they **certainly seem interesting**. Is the inhibition of CSF1 a direct consequence of BRD4 phosphorylation and a general property of all BRD4 inhibitors, or is this a special property of NHWD-870? To what extent does TAM suppression contribute to inhibition of tumor growth?*

In the new **Fig 6**, we showed that CSF1 regulation is through BRD4 and HIF1 α . Therefore CSF1 inhibition is likely the property of all BETi. Please refer to response to reviewer 1 comment #2.

In our previous publication (Yin et al. JCI 2016; 126(11):4157-4173), we showed that TAMs are critical for the growth of ovarian tumors. In this particular setting, it would be hard to separate the effects of TAM suppression and direct tumor suppression in animal models. In **Fig 4h,i**, supplemented TAMs was unable to fully rescue the growth of tumor cells (compare NHWD-870+TAMs vs NHWD-870). However, the lack of full rescue is likely due to lack of proliferating TAMs in the presence of NHWD-870 (**Fig 4j**). In our previous publication (Yin et al. JCI 2016; 126(11):4157-4173), we showed proliferating TAMs express EGF to support the growth of ovarian cancer cells. In any case, we revealed the novel effects of BRD4 inhibition on TAMs, which contribute to tumor growth.

10. The presentation of new data is continued in the Discussion and not much space is dedicated to setting the overall results in perspective.

Thanks for your suggestion. We have removed the data into the main text and added new discussion as suggested.

11. In summary, the manuscript appear to have two main messages, where one is a potent new BRD4

*inhibitor and the other is the observation that the BRD4 inhibitor (and any BRD4 inhibitor?) also suppresses TAM and that this effect is likely to contribute to the anti-cancer activity of the compound. It is **clear** that NHWD-870 is potent, efficacious and interesting. It is also closely related to BMS-986158. Although several pieces of data are included showing that NHWD-870 is somewhat or slightly better than BMS-986158, it is not clear to me that NHWD-870 overall is significantly better. Importantly, unambiguous documentation of purity and identity should be included. The compound certainly deserves to be published, but the data presented here does in my opinion not warrant publication in *Nature Communication*. It is certainly possible that an expansion of the observation of TAM suppression and its link to tumor suppression of BRD4 inhibitors would be suitable for *Nature Communication*.*

Thank you for your comments. As suggested, we have expanded the link of TAM suppression with tumor suppression by BRD4 inhibition/loss. In the new **Fig 6**, we showed how BRD4 loss/inhibition suppresses TAMs by decreasing HIF1 α level to inhibit CSF1 secretion by tumor cells (see reply to Reviewer 1 Comment #2). We hope that the added mechanistic studies and animal experiments improved the manuscript significantly.

In summary, we report the discovery and characterization of a novel, potent, and bioavailable BRD4 inhibitor NHWD-870, which is scheduled to enter phase I trial this spring. We further discovered that BRD4 inhibition by NHWD-870 strongly suppressed proliferation of TAMs mainly by suppressing CSF1 secretion by tumor cells. To our knowledge, we reported for the first time that BRD4 inhibition blocks tumor cell-macrophage interaction. Mechanistically, we showed that BRD4 loss/inhibition suppressed CSF1 expression through downregulating HIF1 α . Thus, our findings provided novel conceptual insight into cancer biology. Together with the report of a novel therapeutic agent, our work will have major clinical impact and be of great interest to broad audience. I hope that you find our revised manuscript suitable for publication in *Nature Communications*. Thank you again for your kind consideration.

Sincerely,

Qin Yan, Ph.D.
Associate Professor
Director, Epigenetics Program
Department of Pathology
Yale School of Medicine

Reviewers' comments:

Reviewer #1 (Remarks to the Author):

The revised manuscript by Yin et al provided some additional data to address the comments made by the reviewers. The major new set of data is in Figure 6 which is intended to demonstrate that CSF1 expression inhibition by compound NHWD-870 is through suppressing BRD4 and HIF1alpha in cancer cells. Other new data include tumor growth inhibition effect by the compound in other models in Figure 3e and 3f, FACS analysis of TAM cells and the exogenous TAM rescue experiments in Supplementary Figure S10, and IHC analysis of tumor CSF1 expression shown in Supplementary Figure S13. Overall, the newly presented data are in good quality. However, the revised manuscript still lacks conclusive information that is sufficient in depth, significance and novelty on the mechanism of action of the new compound and that will be of great interest to a broad readership.

Major comments:

(1). Although the data in Figure 6a and 6b showing that BRD4 knockout or inhibition by the compound strongly decreased HIF1alpha protein expression are intriguing, the manuscript did not provide any further mechanistic information on how BRD4 plays such important role and how the compound affects such function of BRD4. Is it direct, for example, through protein-protein interaction(s)? Or is it indirect through other factors? Does this regulatory function of BRD4 take place at chromatin or off chromatin?

(2) It is also unclear how significant the compound effects on HIF1alpha protein is in its overall anti-tumor activity, at this point. Can those effects be observed in other relevant models? The data presented were mainly obtained from the melanoma cell line A375 or from HeLa cells. The compound inhibition effect on A375 tumors is actually moderate as shown in Figure 3e, when compared to the effects seen with other models such as H526 and A2780. HeLa cells were used only in Figure 6 to show the compound effect on CSF1 expression and the role of HIF1alpha. Does BRD4 play a similar role in control of HIF1alpha protein expression in H526 and A2780? Does compound NHWD-870 exert a similar effect on Hif1alpha in those cell and tumor models? Do other BETi compounds display similar or different activities? Data addressing these questions and the others in (1) will be needed to provide truly new and in-depth information on the mechanism of action of the compound.

(3). It is still unclear in the revised manuscript how (to what extent) the suppression of CSF1 expression by the new compound contributes to the overall tumor inhibition potency. In the manuscript, different cell lines or tumor models are often used to provide data in different assays or analyses. For instance, xenograft tumors with SCLC cell line A526 and large B lymphoma cell line TMD-8 were used in Figure 3 to show the high potency of NHWD-870 in tumor inhibition. But the cell lines were not used in later mechanistic analyses in Figure 5 and 6. As mentioned in (1), the melanoma cell line A375 was used for showing the effect on Hif1alpha protein. It is thus difficult to make an association of CSF1 suppression by the compound with the tumor inhibition potency of the compound. One approach that can be taken to address this is to examine whether ectopic CSF1 expression can strongly mitigate the tumor inhibition effect of the compound.

Other comments:

A. It is unclear how the IHC staining for the three different proteins was scored and normalized.

Reviewer #3 (Remarks to the Author):

I appreciate the effort the authors have made to strengthen their manuscript.

The problem remains that NHWD-870 in many experiments is compared with JQ1 rather than the more closely related BMS-986158, especially in Fig. 2. Where a comparison is included, NHWD-870 tends to show higher potency, especially for the added A375 melanoma cell line. In Fig. 3a, it is a bit unclear which groups the single significance star refers to, as the distance corresponds best to BMS-986158, 3mg/kg, QD and NHWD-870, 1.5mg/kg, BID, but the intended comparison is probably NHWD-870, 3mg/kg, QD?

I agree with the authors that it is a good idea to switch from Stille to Suzuki going forward. However, they don't answer the question regarding if organotin impurities could have influenced the presented results. Such compounds are indeed known to have anticancer properties (see e.g. Devi & Yadav, *Anticancer Agents Med Chem.* 2018;18(3):335-353. doi: 10.2174/1871520617666171106125114; Bulatovic et al, *Angew Chem Int Ed Engl.* 2014 Jun 2;53(23):5982-7. doi: 10.1002/anie.201400763), and it is difficult to exclude that trace impurities could act synergistically with a BET inhibitor. A new synthesis of NHWD-870 with a Suzuki instead of Stille coupling is included, but it is unclear if the new batch has been used for any experiments.

Clearly, all the original experiments were performed with the original batch from the Stille coupling. It would be reassuring with a demonstration that the compounds have not contained organotin impurities or that NHWD-870 synthesized by the Suzuki route has properties that are identical with the previous batch.

The results from keratinocyte viability does not seem to be included as supplementary information to the revised manuscript. Since the authors agree that this is an important question, they probably agree that also other could be interested in this?

It seems likely on basis of the presented data that NHWD-870 represents an advance over BMS-986158 in terms of potency, but apart from with the A375 melanocyte cell line, the advance seems relatively moderate. With such similar compounds, the pharmacokinetic properties and ability to accumulate at the desired site is often more important. Figure S6d (prev. Fig. 3d) in the previous version of this manuscript indicates that BMS-986158 might have a moderate advantage in this respect. For the report of an improved analogue with the same mechanism of action as a previous clinical candidate, I believe a journal such as J. Med. Chem. could be more suitable. Acceptance in Nature Communication would in my opinion depend on the novelty of the mechanistic link between BETi and disruption of communication with tumor-associated macrophages, and clear evidence that the compound acts only by inhibition of BRD4. The link between BETi and HIF-1a seems to have been previously established.

Reviewer #4 (Peplacement of original ref#2, Remarks to the Author):

The authors largely answered to the reviewers' comments by performing additional experiments and providing additional clarifications.

I nevertheless feel that it is important to clearly state in the manuscript that the working mechanism of NHWD-870 appears to be multifactorial, encompassing a direct anti-proliferative effect on macrophages, and an indirect effect on cancer cells which may go beyond a reduction in CSF1 production.

1) add reviewer Figure 2 to the manuscript. The data on BRD4-KO are important as they proof that the absence of BRD4 activity has a direct and significant impact on macrophage proliferation

2) The data in Figure 5n show that CSF1 administration can not rescue macrophage proliferation that have been cocultured with NHWD-870 cancer cells to the same extent as macrophages that have been cocultured with DMSO cancer cells. This suggests that other mechanisms, besides a downregulation of CSF1 production by cancer cells, account for the reduced macrophage proliferation by NHWD-870-treated cancer cells.

Yale University

Mailing address:
Department of Pathology
Yale School of Medicine
P.O. Box 208023
New Haven, CT 06520-8023
TEL: (203) 785-6672
FAX: (203) 785-2443

Shipping address:
Department of Pathology
Yale School of Medicine
310 Cedar Street, BML 348C
New Haven, CT 06510-3128

Email: qin.yan@yale.edu

September 24, 2019

Dear reviewers,

Thank you for carefully reviewing our revised Nature Communications manuscript entitled “Potent BRD4 inhibitor suppresses cancer cell-macrophage interaction”. We are excited that you found our revised manuscript significantly improved and of considerable potential interest, and provided a number of constructive and insightful comments to further improve our manuscript.

To address your comments, we have conducted extensive experiments presented in **Figures. 6e-m, 7b, 7d-i, S15a, e, and Reviewer Figure 1** described below. To summarize, the main experiments included in the revised manuscript are: (1) Demonstrating the *in vivo* role of CSF1 in promoting ovarian cancer growth through regulating macrophages (**Fig 6e-j**); (2) **Demonstrating the ability of CSF1 to strongly mitigate the potent tumor suppressive effects of NHWD-870 (Fig 6k-m)**; (3) Validating the BRD4-HIF1 α -CSF1 axis in A2780 cells (**Fig 7b, d, e, f, i**) (4) validating the regulation of HIF1 α by a different BET inhibitor OTX-015 (**Fig S15a**); (4) Including ChIP-seq data showing that BRD4 binds directly to the HIF1 α promoter and BET inhibitor JQ1 decreases BRD4 binding at the HIF1 α promoter (**Fig 7 g-h**); (5) Directly comparing of NHWD-870 synthesized using Suzuki and Stille coupling, showing that these compounds have similar potency and no side effects in animal experiments (**Reviewer Fig 1**). These new data directly addressed the major concerns by adding additional novel biological mechanistic insights and providing new data to substantiate our conclusions. Detailed information can be found in our point-to-point response to all three reviewers.

We also modified the text according to the reviewers’ constructive suggestions and critiques. To facilitate your review of the revised manuscript, we marked the major change of the manuscript in **red**. Please see below for our point-by-point response to all reviewers’ comments in *italics* below. Again, thank you for your time, effort and very helpful comments, which have helped us to improve our paper.

Reviewer #1 (Expertise: BET inhibitors, cancer):

*1. The revised manuscript by Yin et al provided some additional data to address the comments made by the reviewers. The major new set of data is in Figure 6 which is intended to demonstrate that CSF1 expression inhibition by compound NHWD-870 is through suppressing BRD4 and HIF1 α in cancer cells. Other new data include tumor growth inhibition effect by the compound in other models in Figure 3e and 3f, FACS analysis of TAM cells and the exogenous TAM rescue experiments in Supplementary Figure S10, and IHC analysis of tumor CSF1 expression shown in Supplementary Figure S13. Overall, the newly presented data are **in good quality**.*

We thank the reviewer for the encouraging comments.

2. However, the revised manuscript still lacks conclusive information that is sufficient in depth, significance and novelty on the mechanism of action of the new compound and that will be of great interest to a broad readership.

Thanks for your comments. As described in the summary and below, we provided new results to further substantiate our findings, and we hope the reviewer agree that the newly added information provided additional significance and novelty for a broad readership.

3. Major comments:

(1). Although the data in Figure 6a and 6b showing that BRD4 knockout or inhibition by the compound strongly decreased HIF1 α protein expression are intriguing, the manuscript did not provide any further mechanistic information on how BRD4 plays such important role and how the compound affects such function of BRD4. Is it direct, for example, through protein-protein interaction(s)? Or is it indirect through other factors? Does this regulatory function of BRD4 take place at chromatin or off chromatin?

Consistent with what we proposed in our working model in Fig 7j, HIF1 α is directly regulated by BRD4 at the chromatin. In the new Fig 7g, we found a strong BRD4 binding peaks at the promoter of HIF1 α . In the new Fig 7h, BRD4 binding at the promoter of HIF1 α decreased significantly after BETi inhibitor JQ1 treatment.

4. (2) It is also unclear how significant the compound effects on HIF1 α protein is in its overall anti-tumor activity, at this point. Can those effects be observed in other relevant models? The data presented were mainly obtained from the melanoma cell line A375 or from HeLa cells. The compound inhibition effect on A375 tumors is actually moderate as shown in Figure 3e, when compared to the effects seen with other models such as H526 and A2780. HeLa cells were used only in Figure 6 to show the compound effect on CSF1 expression and the role of HIF1 α . Does BRD4 play a similar role in control of HIF1 α protein expression in H526 and A2780? Does compound NHWD-870 exert a similar effect on Hif1 α in those cell and tumor models? Do other BETi compounds display similar or different activities? Data addressing these questions and the others in (1) will be needed to provide truly new and in-depth information on the mechanism of action of the compound.

Thanks for pointing this out. We validate our findings in A375 and HeLa cells with A2780 model. In the new Fig. 7b, d, e, f, we showed that BRD4 KO or NHWD-870 treatment suppressed HIF1 α level, HIF1 reporter activity and CSF1 expression. In the new Fig. 7i, S15e, we showed that HIF1 α overexpression rescue the CSF1 downregulation by BRD4 knockout.

Different BETi compounds display similar activity on HIF1 α , consistent with idea that this is mediated by BRD4. In the new Fig S15a, a different BETi OTX-015 also decreased HIF1 reporter activity in both A2780 and A375 cells. Consistently, in the new Fig 7h, another different BETi inhibitor JQ1 decreased BRD4 binding at the HIF1 α promoter.

5. (3). It is still unclear in the revised manuscript how (to what extent) the suppression of CSF1 expression by the new compound contributes to the overall tumor inhibition potency. In the manuscript, different cell lines or tumor models are often used to provide data in different assays or analyses. For instance, xenograft tumors with SCLC cell line A526 and large B lymphoma cell line TMD-8 were used in Figure 3 to show the high potency of NHWD-870 in tumor inhibition. But the cell lines were not used in later mechanistic analyses in Figure 5 and 6. As mentioned in (1), the melanoma cell line A375 was used for showing the effect on Hif1 α protein. It is thus difficult to make an association of CSF1 suppression by the compound with the tumor inhibition potency of the compound. One approach that

can be taken to address this is to examine whether ectopic CSF1 expression can strongly mitigate the tumor inhibition effect of the compound.

Thanks for your suggestion to include results with consistent models. After we added the new experiments, we now included all the major experiments using A2780 cells. As the reviewer pointed out, most of the results were validated in other models. For example, we used eight different animal models as shown in **Fig 3** and **S9** to demonstrate the robust effects of the compounds.

To show that CSF1 is critical for macrophage growth, we showed in **Fig 6b-c** that CSF1 is sufficient to promote the growth of macrophages. In the new **Fig 6 e-j**, we showed that ectopic CSF1 expression can strongly promote the growth of A2780 tumors, and its effect is mediated through macrophages, as demonstrated by LC depletion of macrophage. More importantly, as suggested by the reviewer, in the new **Fig 6k-m**, we demonstrated that ectopic CSF1 expression strongly mitigated the tumor inhibitory effects of NHWD-870 in A2780 model.

6. Other comments: It is unclear how the IHC staining for the three different proteins was scored and normalized.

Thank you for pointing this out. We apologize for omitting the information, and have added text to clarify the method of quantification and normalization in the revised manuscript.

Reviewer #3 (Expertise: Medicinal chemistry/pharmacology):

1. I appreciate the effort the authors have made to strengthen their manuscript.

We thank the reviewer for the encouraging comment.

2. The problem remains that NHWD-870 in many experiments is compared with JQ1 rather than the more closely related BMS-986158, especially in Fig. 2. Where a comparison is included, NHWD-870 tends to show higher potency, especially for the added A375 melanoma cell line. In Fig. 3a, it is a bit unclear which groups the single significance star refers to, as the distance corresponds best to BMS-986158, 3mg/kg, QD and NHWD-870, 1.5mg/kg, BID, but the intended comparison is probably NHWD-870, 3mg/kg, QD?

We thank the reviewer for pointing this out. The focus of this paper is NHWD-870 and not BMS-986158, despite that they are related but distinct drugs. The limited comparison showed that NHWD-870 has some advantage against BMS-986158. In addition, no paper has been published on both compounds. Therefore, even data on the clinical stage compound BMS-986158 is novel.

The reviewer was correct that comparison in Fig 3a should be BMS-986158, 3mg/kg, QD vs NHWD-870, 3mg/kg, QD. We apologize not showing clear comparisons. In the revised **Fig 3a**, we moved the lines close to the curves to make it more obvious. Our results showed that NHWD-870, 3mg/kg, QD is superior than BMS-986158, 3mg/kg, QD. In addition, we also compared NHWD-870, 1,5mg/kg, BID with BMS-986158, 3mg/kg, QD and showed that NHWD-870, 1,5mg/kg, BID is similarly superior than BMS-986158, 3mg/kg, QD.

3. I agree with the authors that it is a good idea to switch from Stille to Suzuki going forward. However, they don't answer the question regarding if organotin impurities could have influenced the presented results. Such compounds are indeed known to have anticancer properties (see e.g. Devi & Yadav,

Anticancer Agents Med Chem. 2018;18(3):335-353. doi: 10.2174/1871520617666171106125114; Bulatovic et al, *Angew Chem Int Ed Engl.* 2014 Jun 2;53(23):5982-7. doi: 10.1002/anie.201400763), and it is difficult to exclude that trace impurities could act synergistically with a BET inhibitor. A new synthesis of NHWD-870 with a Suzuki instead of Stille coupling is included, but it is unclear if the new batch has been used for any experiments. Clearly, all the original experiments were performed with the original batch from the Stille coupling. It would be reassuring with a demonstration that the compounds have not contained organotin impurities or that NHWD-870 synthesized by the Suzuki route has properties that are identical with the previous batch.

We again thank the reviewer for suggesting the Suzuki coupling pathway. To exclude the organotin impurities after Stille coupling, we carefully purified the compound by 3 cycles of distillation to remove the organotin compound. We apologize for not clearly labeling the experiments using compounds synthesized using Suzuki coupling method. In the revised manuscript, we modified the text to clearly mark the animal models in **Fig 3d-f** were tested using NHWD-870 synthesized using Suzuki coupling method.

As suggested by the reviewer, to further exclude the effect from organotin impurities, we compared the activity of NHWD-870 synthesized using Stille coupling route with the NHWD-870 synthesized from Suzuki coupling route (**Reviewer Figure 1**). These experiments showed that their activity and possible toxicity were identical.

Reviewer Figure 1 NHWD-870 synthesized through Stille or Suzuki route has similar properties in vivo. Tumor bearing mice were treated with 2mg/kg of the indicated compounds, orally administrated once daily for 21 days. Shown are tumor growth curves (**a**) and body weight change (**b**) for A375 melanoma bearing mice. Data are presented as mean \pm SEM (n=5). n.s., no significant; ***, p<0.001.

4. The results from keratinocyte viability does not seem to be included as supplementary information to the revised manuscript. Since the authors agree that this is an important question, they probably agree that also other could be interested in this?

We agree. As suggested the reviewer, we have added this figure as the new Fig S4.

5. It seems likely on basis of the presented data that NHWD-870 represents an advance over BMS-986158 in terms of potency, but apart from with the A375 melanocyte cell line, the advance seems relatively moderate. With such similar compounds, the pharmacokinetic properties and ability to accumulate at the desired site is often more important. Figure S6d (prev. Fig. 3d) in the previous

version of this manuscript indicates that BMS-986158 might have a moderate advantage in this respect. For the report of an improved analogue with the same mechanism of action as a previous clinical candidate, I believe a journal such as J. Med. Chem. could be more suitable. Acceptance in Nature Communication would in my opinion depend on the novelty of the mechanistic link between BETi and disruption of communication with tumor-associated macrophages, and clear evidence that the compound acts only by inhibition of BRD4. The link between BETi and HIF-1 α seems to have been previously established.

We respectfully disagree with the assessment about the novelty of NHWD-870 as even BMS-986158 has not been published, thus our results characterizing these compounds are novel and will have major impact to direct these compounds for clinical applications. Despite the possible advantage of BMS-986158 in accumulation in the tumor, our comparison of NHWD-870 with BMS-986158 in **Fig 1e, f, 3a, e, S8c, Table S4** and showed the 3-fold increased potency of NHWD-870 led to moderate advantage on suppressing tumor growth and less toxicity.

In addition to characterization of these compounds, we included novel mechanism by which BETi suppresses CSF1 induced macrophage proliferation through HIF1 α . Moreover, although BETi was linked to HIF1 α recruitment to its target gene, we are the first to show that BETi or BRD4 deletion decreased HIF1 α expression, which led to downregulation of its target genes, including CSF1. We have included in the discussion section about the novelty of our results from previous findings. In our response to question #5 of Reviewer #1, we elaborated the importance of CSF1 downregulation to the tumor suppressive effects of BETi with new experiments in **Fig 6**, highlighting the importance of tumor cell-macrophage communication. Taken together, our results provided novel mechanistic insights.

Reviewer #4 (Replacement of original ref#2):

1. *The authors largely answered to the reviewers' comments by performing additional experiments and providing additional clarifications.*

We thank the new reviewer for your assessment.

2. *I nevertheless feel that it is important to clearly state in the manuscript that the working mechanism of NHWD-870 appears to be multifactorial, encompassing a direct anti-proliferative effect on macrophages, and an indirect effect on cancer cells which may go beyond a reduction in CSF1 production.*

1) add reviewer Figure 2 to the manuscript. The data on BRD4-KO are important as they proof that the absence of BRD4 activity has a direct and significant impact on macrophage proliferation

2) The data in Figure 5n show that CSF1 administration can not rescue macrophage proliferation that have been cocultured with NHWD-870 cancer cells to the same extent as macrophages that have been cocultured with DMSO cancer cells. This suggests that other mechanisms, besides a downregulation of CSF1 production by cancer cells, account for the reduced macrophage proliferation by NHWD-870-treated cancer cells.

We agree with the reviewer that although CSF1 plays major roles, other mechanisms including the direct effects of NHWD-870 on macrophages also play important roles. As suggested by the

reviewer, we highlighted the multifactorial mechanism into the revised manuscript into the abstract and description of new **Fig 6**, and added previous Reviewer Figure 2 as the new **Fig S14**.

In summary, we report the discovery and characterization of a novel, potent, and bioavailable BRD4 inhibitor NHWD-870, which will phase I clinical trial soon. We further discovered that BRD4 inhibition by NHWD-870 strongly suppressed proliferation of TAMs partly by suppressing CSF1 secretion by tumor cells. To our knowledge, we reported for the first time that BRD4 inhibition blocks tumor cell-macrophage interaction. Mechanistically, we showed that BRD4 loss/inhibition suppressed CSF1 expression through downregulating HIF1 α . Thus, our findings provided novel conceptual insight into cancer biology. Together with the report of a novel therapeutic agent, our work will have major clinical impact and be of great interest to broad audience. I hope that you find our revised manuscript suitable for publication in *Nature Communications*. Thank you again for your kind consideration.

Sincerely,

Qin Yan, Ph.D.
Associate Professor
Director, Epigenetics Program
Department of Pathology
Yale School of Medicine

Reviewers' comments:

Reviewer #1 (Remarks to the Author):

The re-revised manuscript by Yin et al provided very little additional data to address the comments made by this reviewer. Unfortunately, this re-revised manuscript still lacks the critical data that provide new insights on the mechanism of action (MOA) of the new compound, which can be considered substantial or sufficient in depth, significance and novelty. Therefore, this re-revised manuscript is better suited for journals such as JMC, not for Nat Comm.

Major comments:

(1). The questions raised previously on the MOA of the new compound effect on HIF1alpha expression and MOA at the chromatin level in general are not addressed or addressed sufficiently.

For example, the following questions were asked: 1) how the compound affects such function of BRD4. Is it direct, for example, through protein-protein interaction(s)? Or is it indirect through other factors? Does this regulatory function of BRD4 take place at chromatin or off chromatin?

The only data provided were simple displays at Fig. 7g and 7h of published (by others), anti-BRD4 ChIP-seq peak tracks at the HIF1A gene locus. Fig. 7g displays that the sequencing reads peak slightly higher at the gene promoter region in the A375 melanoma cells subject to ChIP with anti-BRD4 antibody than the reads peak in the cells with input DNA. Based on the data displayed and without other information provided, it is unclear whether BRD4 truly binds to the promoter region of HIF1A. Many antibodies tend to produce ChIP-seq reads that are slightly higher at open chromatin regions such as promoter than the other regions of the chromatin. Fig. 7h displays the ChIP-seq reads at the HIF1A locus in a different cancer cell line (MDA-MB231 breast cancer cells). Overall, the reads displayed appear close to the background reads of the ChIP-seq. The slightly higher reads display at the promoter is again difficult to evaluate for its significance in indication of actual BRD4 binding. Moreover, the cells were treated with BRD4 inhibitor JQ1, but by the author's new compound in this manuscript. Therefore, together, the information provided is very limited in addressing the major questions raised on the MOA of the new compound.

(2). It is puzzling why the authors did not even show data on the compound effect on the mRNA of HIF1A. All the data shown so far are inhibition of HIF1alpha protein expression, which left the reviewers wondering whether the compound acts through transcriptional regulation of HIF1A gene expression. This is why the question was asked about whether the effect is direct or indirect.

(3). Overall, to address the comments directly on the MOA of the new compound, the authors should perform experiments such as CHIP-seq with antibodies against BRD4 and some of the relevant histone marks with cells or tumors treated with the author's new compound and other compounds such as JQ1 or other ones from pharmaceutical industry and compare their actual actions on the chromatin levels.

Reviewer #3 (Remarks to the Author):

The revised manuscript has strengthened the case that BRD4 inhibition counteracts tumors by inhibition of TAM via HIF1 and CSF1. The authors have satisfactorily addressed all my points in the revised manuscript and the rebuttal.

Reviewer #4 (Remarks to the Author):

The authors addressed my concerns

Yale University

Mailing address:
Department of Pathology
Yale School of Medicine
P.O. Box 208023
New Haven, CT 06520-8023
TEL: (203) 785-6672
FAX: (203) 785-2443

Shipping address:
Department of Pathology
Yale School of Medicine
310 Cedar Street, BML 348C
New Haven, CT 06510-3128
Email: qin.yan@yale.edu

January 15, 2020

Dear reviewers,

Thank you for carefully reviewing our revised *Nature Communications* manuscript entitled “Potent BRD4 inhibitor suppresses cancer cell-macrophage interaction”. We are excited that Reviewers #3 and #4 were completely satisfied with our revision.

To address the remaining concerns of Reviewer #1, we have conducted additional experiments. Specifically, we have now shown that NHWD-870 decreases HIF1A mRNA levels (new **Fig 7g**), and BRD4 binds to the HIF1A promoter and NHWD-870 blocks BRD4 binding to the HIF1A promoter (new **Fig 7i**).

To facilitate your review of the revised manuscript, we marked the major change of the manuscript in **red**. Please see below for our point-by-point response to Reviewer #1’ comments in *italics* below. Again, thank you for your time, effort and very helpful comments, which have helped us to improve our paper.

Reviewer #1 (Expertise: BET inhibitors, cancer):

1. The re-revised manuscript by Yin et al provided very little additional data to address the comments made by this reviewer. Unfortunately, this re-revised manuscript still lacks the critical data that provide new insights on the mechanism of action (MOA) of the new compound, which can be considered substantial or sufficient in depth, significance and novelty. Therefore, this re-revised manuscript is better suited for journals such as JMC, not for Nat Comm.

We respectfully disagree with the assessment. We would like to point out that we have performed significantly amount of experiments that directly address your comments in our last revision.

1. To show that CSF1 is critical for macrophage growth, we showed in **Fig 6b-c** that CSF1 is sufficient to promote the growth of macrophages.
2. In the **Fig 6 e-j**, we showed that ectopic CSF1 expression can strongly promote the growth of A2780 tumors, and its effect is mediated through macrophages, as demonstrated by LC depletion of macrophage.
3. In the new **Fig 6k-m**, we demonstrated that ectopic CSF1 expression strongly mitigated the tumor inhibitory effects of NHWD-870 in A2780 model.
4. We have included all the major experiments using A2780 cells and validated most of the results in other models. For example, we used eight different animal models as shown in **Fig 3** and **S9** to demonstrate the robust effects of the compounds. In **Fig. 7b, d, e, f**, we showed that BRD4 KO or NHWD-870 treatment suppressed HIF1 α level, HIF1 reporter activity and CSF1 expression.

In the **Fig. 7j, S15e**, we showed that HIF1 α overexpression rescue the *CSF1* downregulation by BRD4 knockout.

5. We showed that different BETi compounds display similar activity on HIF1 α , consistent with idea that this is mediated by BRD4. For example, in the **Fig S15a**, a different BETi OTX-015 also decreased HIF1 reporter activity in both A2780 and A375 cells.

2. The questions raised previously on the MOA of the new compound effect on HIF1alpha expression and MOA at the chromatin level in general are not addressed or addressed sufficiently.

For example, the following questions were asked: 1) how the compound affects such function of BRD4. Is it direct, for example, through protein-protein interaction(s)? Or is it indirect through other factors? Does this regulatory function of BRD4 take place at chromatin or off chromatin?

The only data provided were simple displays at Fig. 7g and 7h of published (by others), anti-BRD4 ChIP-seq peak tracks at the HIF1A gene locus. Fig. 7g displays that the sequencing reads peak slightly higher at the gene promoter region in the A375 melanoma cells subject to ChIP with anti-BRD4 antibody than the reads peak in the cells with input DNA. Based on the data displayed and without other information provided, it is unclear whether BRD4 truly binds to the promoter region of HIF1A. Many antibodies tend to produce ChIP-seq reads that are slightly higher at open chromatin regions such as promoter than the other regions of the chromatin. Fig. 7h displays the ChIP-seq reads at the HIF1A locus in a different cancer cell line (MDA-MB231 breast cancer cells). Overall, the reads displayed appear close to the background reads of the ChIP-seq. The slightly higher reads display at the promoter is again difficult to evaluate for its significance in indication of actual BRD4 binding.

Moreover, the cells were treated with BRD4 inhibitor JQ1, but by the author's new compound in this manuscript. Therefore, together, the information provided is very limited in addressing the major questions raised on the MOA of the new compound.

Thanks for your comments. To further address your concern, we have performed ChIP-qPCR analyses of A2780 cells and found that BRD4 binds to the HIF1A promoter and NHWD-870 treatment abolishes BRD4 binding to the HIF1A promoter. These results in the new **Fig 7i** are consistent with our working model in Fig 7k, HIF1 α is directly regulated by BRD4 at the chromatin.

3. It is puzzling why the authors did not even show data on the compound effect on the mRNA of HIF1A. All the data shown so far are inhibition of HIF1alpha protein expression, which left the reviewers wondering whether the compound acts through transcriptional regulation of HIF1A gene expression. This is why the question was asked about whether the effect is direct or indirect.

Thanks for your important comments. We now include the RT-PCR analyses as suggested. NHWD-870 indeed decreased HIF1A mRNA levels in new **Fig 7g**.

4. Overall, to address the comments directly on the MOA of the new compound, the authors should perform experiments such as ChIP-seq with antibodies against BRD4 and some of the relevant histone marks with cells or tumors treated with the author's new compound and other compounds such as JQ1 or other ones from pharmaceutical industry and compare their actual actions on the chromatin levels.

Thanks for your suggestion. In our response to your comment #1, we have shown with ChIP-qPCR that NHWD-870 suppresses BRD4 binding to the *HIF1A* promoter. Further extensive ChIP-seq

experiments could reveal additional mechanisms of action, but we feel beyond the scope of the current study.

In summary, we report the discovery and characterization of a novel, potent, and bioavailable BRD4 inhibitor NHWD-870, which will enter Phase I clinical trial soon. We further discovered that BRD4 inhibition by NHWD-870 strongly suppressed proliferation of TAMs partly by suppressing CSF1 secretion by tumor cells. To our knowledge, we reported for the first time that BRD4 inhibition blocks tumor cell-macrophage interaction. Mechanistically, we showed that BRD4 loss/inhibition suppressed CSF1 expression through downregulating HIF1 α . Thus, our findings provided novel conceptual insight into cancer biology. Together with the report of a novel therapeutic agent, our work will have major clinical impact and be of great interest to broad audience. We believe that you will find our revised manuscript suitable for publication in *Nature Communications*. Thank you again.

Sincerely,

Qin Yan, Ph.D.
Associate Professor
Director, Epigenetics Program
Department of Pathology
Yale School of Medicine